# Secretory System Components as Potential Prophylactic Targets for Bacterial Pathogens

**DOI:** 10.3390/biom11060892

**Published:** 2021-06-15

**Authors:** Wieslaw Swietnicki

**Affiliations:** Department of Immunology of Infectious Diseases, Hirszfeld Institute of Immunology and Experimental Therapy, Polish Academy of Sciences, ul. R. Weigla 12, 53-114 Wroclaw, Poland; wieslaw.swietnicki@hirszfeld.pl

**Keywords:** vaccine, secretory system, bacteria, *Yersinia pestis*, *Salmonella enterica*, pathogenic *Escherichia coli*, *Pseudomonas aeruginosa*, *Shigella flexneri*

## Abstract

Bacterial secretory systems are essential for virulence in human pathogens. The systems have become a target of alternative antibacterial strategies based on small molecules and antibodies. Strategies to use components of the systems to design prophylactics have been less publicized despite vaccines being the preferred solution to dealing with bacterial infections. In the current review, strategies to design vaccines against selected pathogens are presented and connected to the biology of the system. The examples are given for *Y. pestis*, *S. enterica*, *B. anthracis*, *S. flexneri*, and other human pathogens, and discussed in terms of effectiveness and long-term protection.

## 1. Introduction

Strategies to deal with bacterial infections can be divided into two major groups: therapeutics, and prophylactics. The former relies on antibiotics, mostly, while the latter- predominantly on vaccines. Antibiotics-based strategies are becoming a major problem as the pathogens acquire resistance faster than the typical approval process for new drugs [1]. The situation is caused by economical disadvantages for the pharmaceutical companies to invest in new drugs when the typical therapy lasts only a week [2,3,4]. To remedy low revenues in this area, the strategy uses existing targets with minimal modifications of the basic drug scaffold [5]. Predictably, resistance generation among bacteria is also shortened and frequently involves small modifications of the same bacterial targets.

The existing antibiotic pipelines are based on 2 major targets: peptidoglycan biosynthesis and DNA replication as proven over time [6,7]. While the first target generates relatively small side effects, the second one is frequently problematic and not recommended in many cases [8,9]. The selection of new targets and drugs is encouraged and funded by the public-private partnerships [10] but the speed at which it progresses is slow due to the inadequate funding, the economic viability of antibiotic research for pharmaceutical companies, legal requirements connected to the approval of new drugs, and the rapid rise of resistance fueled by massive and indiscriminate, frequently, antibiotics use in humans and farming [11,12]. The situation is critical and the corrective actions undertaken are not sufficient to prevent major drug resistance epidemics in the future [13]. Therefore, approaches favoring alternatives [14,15], including indirect strategies relying on blocking bacterial virulence systems [16], are becoming more prominent and receiving increased funding from the same public-private partnerships [17,18].

Preventive strategies for bacterial infections are favored in the long term as it is easier and cheaper to prevent the disease than to treat it in many cases. Vaccine design and selection are typically based on attenuated strains, inactivated pathogens (disfavored due to the complications), and acellular approaches relying frequently on selected bacterial proteins fused to bacterial lipopolysaccharides as adjuvants [19]. However, the selection process is a trial-and-error approach with the majority of candidates failing in the clinical trials [20]. For many prophylactics, the vaccine design strategies did not account for the complexity of the infection process of the pathogen and did not always select animal models capable of fully reproducing the interaction process with the human host [21,22]. The strategy for bacterial pathogens should be optimized in the future. The recent response to the COVID-19 viral pandemic demonstrated a clear focus on functional vaccine delivery by combining funding, scientific knowledge, animal models, manufacturing, and logistics to deliver a set of viable candidates for human use, possibly making it an example of future work on bacterial prophylactics.

Bacterial pathogens all have specialized protein transport systems and they are cataloged at the Kyoto Encyclopedias of Genes and Genomes (KEGG) database (https://www.genome.jp/kegg/). The infection process requires not only finding a receptor on the human host but also the ability to secrete protein components in a coordinated process to overcome the immune defenses of the host and establish a suitable niche to replicate after infection. Theoretically, blocking the bacterial transport by virulence systems would block the infection and render the bacterial pathogens defenseless [23,24,25]. Since the bacterial transport systems are required for bacterial pathogens to start and continue an infection process and can be easily identified bioinformatically from the DNA sequences [26], there is an alternative vaccine design strategy that could be used to prevent bacterial infections. In the current work, such strategies involving bacterial secretory systems as targets are presented for selected pathogens and discussed based on experimental data.

The review concentrates on components of virulence systems as building blocks of vaccines (Table 1). Therefore, information on other vaccine candidates not based on components of virulence systems is substantially shortened.

## 2. Results

The approaches dealing with vaccine strategies for different bacterial pathogens depend on the biology of the organism. It involves its mechanism of infection and the roles of identified protein transport systems in the process.

### 2.1. Acinetobacter baumannii

*Acinetobacter* is an opportunistic Gram-negative pathogen genus. While most strains are non-pathogenic, the *baumannii* species are becoming drug-resistant and problematic under hospital settings. There is no commercial vaccine against this pathogen available despite many years of research [27]. The only therapy is with antibiotics.

#### 2.1.1. Virulence Systems

The *Acinetobacter baumannii* has a type I, II, and type VI secretion system. However, the last system is present only in selected strains and many hospital isolates do not have it [28]. Despite its known target of other bacteria, the system is also important for virulence as its deletion affected the pathogen’s lethality in mice [29].

#### 2.1.2. Vaccines

Existing vaccine candidates range from live attenuated/engineered pathogens [30] to recombinant proteins conjugated to bacterial capsular antigens (review in Gellings et al. [31]). An alternative strategy, relying on the pan-genomic analysis of sequenced genomes, was proposed by Hassan et al., 2016 [32]. The approach used bioinformatics analysis to identify targets based on the conserved core proteins that were further divided into virulence, essential and non-host homolog proteins. Prediction of localization, biological function, potential epitopes, and interaction partners identified 12 candidates, of which only 5 were from secretory systems: pilus assembly protein porin PapC and a pilus assembly chaperone, general secretion pathway protein D (2 candidates), and the type VI secretion system OmpA/MotB protein. The rest of the candidates were membrane proteins involved in transport or cellular structure assembly mostly. The findings will have to be tested in animals to confirm potential applicability in human vaccine designs.

### 2.2. Bacillus anthracis

The pathogen is a Gram-positive bacterium infecting mainly herbivores. Humans are accidental hosts through contact with animals or their products. Due to its destructive potential, the pathogen is classified as the Class A Select Agent by the Center for Disease Control (CDC).

#### 2.2.1. Virulence Systems

The bacterium has 2 main virulence factors, the tripartite toxin Protective Antigen (PA) produced on the plasmid pXO1 [33] and the capsule composed of poly-γ-D-glutamic acid (PDGA) produced on the plasmid pXO2 [34]. The PA forms oligomers, hepta- or octamers when cleaved by furin protease, and forms a pore in cellular membranes for delivery of 2 other toxins: metalloprotease Edema Factor (EF) [35,36] and adenylate cyclase Lethal Factor (LF) [37].

#### 2.2.2. Vaccines

There are 2 vaccines available commercially against the pathogen: the acellular Anthrax Vaccine Precipitated (AVP) consisting of alum-precipitated *B. anthracis* Sterne strain crude culture filtrates [38] and the Anthrax Vaccine Adsorbed (AVA) aluminum hydroxide-adsorbed product consisting mainly of PA from the *B. anthracis* strain V770-NP1-R. Both vaccines have to be given intramuscularly and the injection repeated after a year to maintain their effectiveness.

Cellular vaccines have many side effects and there is a trend to transition towards acellular antigens. Accordingly, Kachura et al. [39] used nanoparticles composed of Ficoll conjugated to CpG oligonucleotides and recombinant PA protein as an antigen. The particles also contained Toll-like receptor 9 (TLR9) ligand DV230 on its surface. The challenge study was performed in non-human primates Cynomolgus monkeys. Animals were immunized with 10 μg of rPA, 250 or 1000 μg of Ficoll-DV230-rPA antigens via i.m. route with a booster on day 29 and challenged with 200 LD50 of aerosolized dose of *B. anthracis* Ames on days 69, 70 and 71. Immunization with the Ficoll-DV230-rPA construct fully protected Cynomolgus macaques from *B. anthracis* infection-induced death after a single (1000 μg) or double (250 μg and 1000 μg) immunization. The result is very promising but the typical background level in vaccine testing is 100–200 LD50.

In a different attempt, a live vaccine was constructed based on a non-pathogenic *B. subtilus* strain displaying PA on its surface [40]. Mice were immunized via i.g. lavage repeatedly (7× in 35 days) with 5 × 10^9^ spores/100 μL in PBS and other routes: i.n., i.p., and s.l., 3× over 28 days using approximately 1 × 10^9^ spores per immunization. The preparations were combined with cholera toxin B subunit as a mucosal delivery vehicle. Animals were challenged with 6 × 10^7^ CFU *B. anthracis* Ames strain (Tox + Cap−) which corresponded to 100 MLD50. Only the animals immunized via the i.p. route were fully protected while immunization via oral and intranasal routes had protection rates of 80% and 90%, respectively, judging by survival rates. The rest of the animals had protection rates of 30% or below.

### 2.3. Bordetella bronchiseptica

The pathogen is a Gram-negative bacterium infecting mainly dogs and pigs. Humans are accidental hosts.

#### 2.3.1. Secretory Systems

The bacterium has type I, II, III, twin-arginine targeting (Tat), and type VI secretion systems according to a KEGG database (https://www.kegg.jp/kegg-bin/show_pathway?bbm03070). Recognized virulence factors include haemagglutinins which are typically expressed on the surface of bacteria and secreted [41,42,43,44].

#### 2.3.2. Vaccines

Commercial vaccines for dogs include many products but their efficacy is mainly left in the realm of conjecture due to the lack of a systematic study for larger cohorts [45]. There is, however, a trend to use live strains, mainly *Salmonella*, as an engineered display vector for antigens from the *B. bronchiseptica* bacterium. There are 2 instances described in the review.

In the first one, Hur et al. [46] used an engineered attenuated *Salmonella* vector expressing a fusion of filamentous haemagglutinin (FHA) fragment from *B. bronchiseptica* with pertactin to evaluate its effectiveness in a mouse model of progressive atrophic rhinitis (PAR) caused by *Pasteurella multocida*. The former is commonly observed in infections with the latter. The BALB/c mice were inoculated intranasally with approximately 0.2 × 10^5^ CFU of the vaccine strain. In the work of Hur et al., the animals were partially protected against *P. multocida* infection but the data was too spread to draw more conclusions.

In the second instance, Zhao et al. [47] used a recombinant fusion protein of the same fragments displayed by an attenuated *Salmonella enterica * Cholerasius strain in a mouse model of *B. bronchiseptica* and *S. enterica* infections. Ther BALB/c mice were immunized with 2.1 × 10^8^ CFU (s.c.) or 2.1 × 10^10^ CFU (oral) of the engineered strain twice over the 2 weeks and challenged 30 days later with 5.2 × 10^6^ CFU of the *B. bronchiseptica* HH0809 strain. The s.c. route was the best and all animals survived as opposed to the oral route that rendered only 20% protection. The used challenge dose was equal to approximately 4 LD50 which is much below the 100–200 LD50 level seen as a background in vaccine work.

### 2.4. Bordetella pertussis

The pathogen is a Gram-negative, aerobic bacterium infecting predominantly humans.

#### 2.4.1. Secretory Systems

The species has type I, II (general), III, and Tat secretory systems according to the KEGG database (https://www.kegg.jp/kegg-bin/show_pathway?bpet03070), of which type I and III are involved in virulence [48]. The type I secretion system secretes adenylate cyclase CyaA which is critical in establishing pathogen infection [49]. Other important virulence factors are pertussis toxin (PT), type III secretion effectors BopB, BopD, and Bsp22 [50], and filamentous haemagglutinin (FHA) [51].

#### 2.4.2. Vaccines

There is an acellular commercial vaccine DTaP against the pathogen which replaced the whole cell-based DTwP vaccine (review in [51]). The vaccine consists of 3–5 epitopes as opposed to all epitopes in the DTwP. Analysis of immune responses to DTwP and DTaP vaccines [52] showed that the acellular version had a weak Th1 response while having good Th2 and Th17 responses. However, the whole cell-based DTwP had strong immune responses from Th1 and Th17 cells. To correct the immune response bias, the inclusion of CpG as an adjuvant in the DTaP was shown as a viable solution [52]. However, the adjuvant for the DTaP was chosen as alum-based which normally does not have a good Th1 response. The polarized immune response from the DTaP could likely be one of the problems associated with a resurgence of *B. pertussis* in the population. The other reason for the resurgence may be the limited antigen repertoire of the DTaP vaccine.

To overcome the poor efficacy of the DTaP vaccine, many groups started pursuing live vaccines or outer membrane vesicles (OMVs) strategies as the basis of new vaccine formulations. In the first category, a Swedish group tested clinically [53] the BPZE1 vaccine based on the genetically engineered *B. pertussis* strain in which the major toxins were replaced by enzymatically inactive analogs [54]. A single dose of 10^9^ CFU of the attenuated strain was administered intranasally and analyzed for up 12 months for the type of immune response and the ability to maintain it over time. The live vaccine could elicit a Th1-type response as opposed to the predominantly Th2-type for the acellular vaccine. The antibody repertoire induced by the live strain was also broad as opposed to the narrow one of the acellular vaccine that was limited to the vaccine antigens.

The phenomenon of extracellular vesicles being formed by bacteria has been known [55] but the technology to harvest it is becoming more available only recently. In the work of Gaillard et al. [56], the OMVs from *B. pertussis* mutant expressing the lipid A deacetylase PagL were conjugated to DTaP toxoids and used to immunize BALB/c mice. The used dose was 1.75 ug of total OMV protein/animal in a 2-dose intraperitoneal protocol over 2 weeks and challenged intranasally with a sublethal dose (10^6^–10^8^ CFU/40 μL) of *B. pertussis* Tohama phase I, Bp18323 or Bp106 clinical isolate. The immunization could reduce bacterial load by 5 logs in the lungs compared to the placebo and the immunity was maintained for 5 months with a substantial reduction after 9 months as judged by the bacterial count in the lungs. However, the protection from infection was not determined and the data has to be provided to evaluate the construct in full.

Another strategy, based on nanoparticles, was used by Najminejad et al. [57]. The nanoparticles were encapsulated and contained FHA and PT as a novel vaccine candidate in a mouse model of vaccination. The BALB/c mice were immunized intranasally or intraperitoneally twice over 28 days using a 6 μg of PT and 6 μg FHA combination. The efficacy was evaluated by ELISA of sera antibodies and cytokine profiles but not in an infection model. The response showed that the strategy produced a significant immune response in animals.

### 2.5. Brucella abortus

The pathogen is a Gram-negative bacterium encountered mainly in animals and one of the causative agents of brucellosis. The disease is highly contagious and of significant concern to farmers due to potential economic losses. Treatment of animals is not recommended while humans are treated with different antibiotics combinations [58].

#### Vaccines

There is a live vaccine based on an attenuated S19 strain [59] and it is licensed for animal use. The vaccine, however, can cause abortion in cattle and is not always preferred for this reason [60,61]. A search for a better and acellular vaccine is in progress but no acellular vaccine for *B. pertussis* has been approved for animal use.

Acellular vaccine candidate development was tested by Pollak et al. [62]. The group used VirB7 and VirB9 proteins from the type IV secretion system as antigens in a mouse model of brucellosis. The immunization was performed via the i.p. route with 30 μg of antigen and included a booster shot 15 days post-immunization using the BALB/c mouse strain. The challenge was with the fully virulent *B. abortus* 544 strain (4 × 10^4^ CFU). Protection was quantified by bacterial count in spleens of infected animals. Both VirB7 and VirB9 proteins offered only small protection by reducing bacterial spleen load by a log of magnitude as compared to the placebo, while the attenuated S19 strain offered a reduction by 2 logs of magnitude.

An alternative strategy used interference with the virulence regulatory network Two-Component Regulatory System (TCS) [63] by utilizing a live vaccine *B. abortus* 2308 strain with 2 of the critical genes *nodV* and *nodW* deleted, alone or together [64]. The mutants were highly attenuated and offered protection against the fully virulent 2308 strain in a mouse model of brucellosis. Bacterial spleen counts were reduced by 3 logs and the vaccinated animals could survive the challenge with the fully virulent strain. The animals were cleared of the pathogen by week 10 when using either of the live vaccine strains by counting bacterial spleen loads.

### 2.6. Brucella melitensis

The bacterium is a Gram-negative pathogen and one of the causes of brucellosis in animals. There is a live vaccine licensed for animal use [65] but it still causes abortion in vaccinated animals. Therefore, alternative solutions are sought after. Treatment of animals is not recommended but humans are treated with antibiotics [66].

#### Vaccines

The VirB proteins from the type IV secretion system were not reported tested as potential vaccine candidates against *B. melitensis* infection. Instead, a live mutant defective in LPS biosynthesis, *B. melitensis* 16M*Δwzt*, was used as a potential vaccine candidate [67]. The mutant was defective in the critical wzt protein essential for O-type LPS antigen transport to the surface. In a mouse infection model with *B. melitensis*, the BALB/c mice were inoculated with 10^6^ CFU of the mutant strain via the i.p. route and challenged with 10^6^ CFU of the wt or M5 strain 16 weeks post-inoculation. Vaccination with the mutant reduced spleen bacterial load by 2 logs when challenged with the wt strain. Approximately the same level of reduction was observed for the challenge with *B. melitensis* 5M stain. The *Δwzt* mutant could be a future vaccine candidate against *B. melitensis*.

### 2.7. Chlamydia trachomatis

The bacterium is a highly obligate intracellular pathogen and the cause of sexually transmitted diseases in women, often leading to fallopian tube closures and infertility. There are 3 major serovar groups, each responsible for different infections. The first group with serovars A-C infects eyes and is responsible for blindness and conjunctivitis [68]. The second group with serovars D-K is responsible for genital infections [69], and the third one, lymphogranuloma venereum biovar, with serovars L1-3- for urogenital and anorectal infections [70]. There is no commercial vaccine available for the pathogen and the only therapy is with the use of antibiotics.

#### 2.7.1. Secretory Systems

The bacterium has a highly conserved type III, pathogenic type II and a general secretory system (https://www.kegg.jp/kegg-bin/show_pathway?ctr03070). The first one is also present in the *Chlamydia muridarum*. The system produces 2 translocators, CopB and CopD, which insert themselves into mammalian membranes and form a receptor for the tip of the needle conduit transporting bacterial effectors. Antibodies to both proteins were detected in the blood of patients infected with *C. trachomatis* [71]. Both *Chlamydia* species also express the highly conserved CT584 protein [72].

#### 2.7.2. Vaccines

The vaccine candidates for *C. trachomatis* have been studied for a long time. The most promising were based on the Major Outer Membrane Protein (MOMP) as the protein is the most expressed bacterial antigen (Review in [73]). It has been recognized as capable of inducing both humoral and cellular immunity [74,75]. However, the most advanced vaccine design is based on the recombinant CTH522 protein that has reached Phase I clinical trials [76].

The use of type III secretion system proteins as antigens in a vaccine study has been reported only once. Bulir et al. [72] used a recombinant fusion of CopB (a.a. 1-100)-CopD (a.a. 1-100)-CT584 as an antigen. C57Bl/6 mice were immunized intranasally with 20 μg of antigen with 10 μg of CpG ODN1826 twice over 3 weeks and then challenged intravaginally with 10^5^ IFU of *C. muridanum*. The candidate reduced the infection rate by 80% as judged by the vaginal bacterial load and the pathology of the oviduct by 90% in the mouse model of *Chlamydia* infection. Bacterial shedding was reduced by 95%, a sign of a potential vaccine candidate for future development.

### 2.8. Pathogenic Eschericchia coli

*Escherichia coli* bacteria may cause diarrhea, among other diseases. The pathogenic strains have a variety of virulence markers but are typically classified into 6 groups based on the presence of loci responsible for those properties [77,78].

#### 2.8.1. Secretory Systems

The enteropathogenic group (EPEC) is defined as producing attaching and effacing (A/E) lesions and lacks genes producing Shiga toxin while the enterohaemorrhagic *E. coli* (EHEC) has a gene producing Shiga toxin. Both groups have a type III secretion system that is tightly regulated [79] and sometimes a second system is also present [80,81]. The other 4 groups are enteroaggregative (EAEC), diffusely aggregative (DAEC), enterotoxigenic (ETEC), and enteroinvasive (EIEC) *E. coli*.

#### 2.8.2. Vaccines

The variety of pathogenic markers and connected with them infection mechanisms [82] make it difficult to have a universal vaccine suitable for all strains. Predictably, there has not been a universal commercial human vaccine available for any of those strains despite many vaccines against individual strains being evaluated clinically [83]. There are, however, commercial vaccines for animals designed to reduce the infection by EHEC [84,85] and other strains [86].

There are 2 approaches to vaccines described in this review that used elements of secretory systems as vaccine candidates. In the first one, the development of a human universal vaccine against pathogenic *E. coli* has been proposed for members of the serogroup O111 which includes EPEC, EAEC, and Shiga toxin-producing strains [78]. The system used LPS conjugated to cytochrome or a recombinant secreted toxin LT from ETEC [87]. Rabbits were immunized via s.c. route using 5 µg/mL of protein-LPS conjugate suspended in a proprietary mix of mineral oil and detergents incorporated in Sılica SBA-15 nanoparticles 6 times within a year to generate antibodies. BALB/c mice were immunized orally with 0.2 mL of O111-EtxB conjugate resuspended in the proprietary formulation 3 times over 30 days. The protection was determined by the ability of generated antibodies to prevent cell agglutination by the end-point dilution method and by the ability to prevent cytopathic effect on the Y1 cell line. The endpoint dilution assay showed about 3 logs of difference between the nonvaccinated and vaccinated animal sera antibodies while the cytopathic assay showed about a log of A405 difference for the same sera. Although the animal protection was not shown, the vaccine candidate showed promising data in generating antibody response in mice for the ETEC when the secreted toxin was used in the conjugate.

In the second approach, Byrd et al. [88] used an attenuated mutant of rabbit EPEC strain with the *ler* virulence operon regulator deleted as a live vaccine. The deletion silences gene expression and makes it an effective live vaccine strain against the isogenic pathogen [89]. The construct in Byrd et al.’s work contained a fusion of EspP autotransporter with the B subunit of Shiga toxin 1 (Stx-1) replacing its passenger domain and was designed to produce the fusion in the cytoplasm, periplasmic space, on the surface, or be secreted. In a rabbit model of EHEC infection, New Zealand White rabbits were immunized via the orogastric route with 10^9^ CFU of the mutants strains and boosted again after 2 weeks. The animals were challenged with 10^9^ CFU of the RDEC-H19 strain after 2 weeks via the same route. Protection was estimated by the stool consistency, weight loss of the animals, and the production of anti-Stx-1 antibodies by ELISA. The construct producing the secreted form of the fusion protein was the most protective with 4 out of 5 animals capable of neutralizing half of the secreted Stx-1 and having a substantially improved stool consistency compared to the rest of the constructs.

### 2.9. Francisella tularensis

The bacterium is a Gram-negative zoonotic intracellular pathogen infecting almost any host and the causative agent of tularemia. The pathogen is a Category A Select Agent due to its combination of infectivity and lethality.

#### Vaccines

There is no commercial vaccine available for humans but there is a live attenuated strain *F. tularensis* subsp. *holarctica* LVS strain being used as a live vaccine [90] for special groups. The strain was originally developed in the former Soviet Union and further optimized in the US [91]. Due to its relative toxicity, there have been attempts to reduce toxicity and increase the protection of the resulting strains. Various mutants, including those within the operon coding for the virulence system, have been used as live vaccines [90,92,93] with a different degree of success. The use of recombinant proteins or isolated cellular components as vaccines failed to deliver a candidate with a strong immune response, either [94,95,96,97].

The live vaccines were left as a potentially viable option to protect against highly virulent *F. tularensis subsp.* tularensis SCHU S4 or FSC033 variants isolated from different sources [91,98]. Mutations of single genes from the type VI secretion system *ΔiglB* [99,100], *ΔiglB::fljb* [101], Fn iglD [102], or the *Francisella* Pathogenicity Island (FPI) promoter regulator *pmrA* [103] failed to deliver live candidates with protection against challenges with LD50 above 100 of the virulent strains.

In an interesting variation of live vaccines, Jia et al. [104] used an attenuated LVS strain with deletion of the capsule protein capB and resupplied with the iglA, B, and C proteins, separate or together, under the control of *F. tularensis bfr* promoter (*ΔcapB/bfr-iglABC*) to protect against the virulent SHU S4 variant of *F. tularensis*. The BALB/c mice were immunized 2–3 times, 4 weeks apart, with the mutant strains at 10^6^–10^7^ CFU and then challenged intranasally with 10 LD50 of the SHU S4 variant. The degree of protection was correlated with the bacterial load used for immunizations and their time. At 10^6^ CFU of the mutant used immunizations performed 2 days before challenge, only 20% of animals could survive the challenge up to 20 days post-challenge, twice the number of animals compared to the higher dose. Change of route to i.d. and the extension of time before the challenge to 49 days demonstrated that the LVS vaccine could protect 50% of the animals from the challenge with 16 CFU of the SHU S4 strain for 20 days. An increase of challenge dose to 31 CFU showed that the best variant was the *bfr-iglA* construct that could protect about 30% of the animals at the same time, in contrast to other constructs where all animals died after that time.

As an alternative to recombinant proteins, secretory vesicles of bacteria are used as antigens. There is only one report of secretory vesicles encapsulated in a cationic detergent being used as a vaccine candidate against tularemia [105]. The vesicles were prepared from the live attenuated LVS or the fully virulent SHU S4 strains. The BALB/c mice were immunized with 35 μg of protein in vesicles up to 3 times in 2-week intervals via i.p., i.n. or s.c. routes and challenged 14 days after the last dose. The animals were challenged with up to 100,000 CFU of the LVS via the i.p. route or 50 CFU of SHU S4 via i.n. route. Challenge with up to 70,000 CFU of the LVS strain resulted in deaths for the LVS vesicle-immunized animals. Passive immunization with 60 µL of serum/animal via tail vein 1 day before challenge with 100,000 CFU of the LVS strain could protect up to 80% of the animals up to 14 days as judged by survival rates.

### 2.10. Helicobacter pylori

The bacterium is a Gram-negative pathogen infecting about half of the world’s human population [106,107,108]. It is also named the number one carcinogen by the WHO due to its association with gastrointestinal pathology, peptic, and gastric ulcers [109]. An animal model showing a direct association between *H. pylori* infection and gastric cancer was developed in Mongolian gerbils by Tatematsu et al. [110].

#### 2.10.1. Secretory Systems

Analysis of genomic DNA showed that the pathogen has general, type Va, Tat, and type IV secretory systems (https://www.kegg.jp/kegg-bin/show_pathway?hhp03070). The last system is associated with pathogenicity and is critical for its virulence in the human host. In prokaryotes, the type IV system is used to transport DNA and large proteins. In *H. pylori*, the pathogen encodes 4 different type IV secretion systems, each with a specific function in infection [111]. The first system, located within the *cag* pathogenicity island (*cag*PAI) and associated with gastric ulcers [112], is used to transport the cytotoxicity-associate gene A (CagA) protein through its pilus (review in [113]). The secreted CagL effector is the pilus receptor that binds to human integrins of the α5β1 family [114], later causing activation of Src tyrosine kinase for CagA phosphorylation. The process leads to multiple changes in the cell, including disruption of adherens and tight junctions, actin depolymerization, changes in membrane dynamics, and pro-inflammatory nuclear transcription. The pathogen also secretes another protein, the vacuolating cytotoxin A (VacA), via an autotransporter mechanism (review in [115]), causing vacuolation of mammalian cells. Protection from stomach acid is rendered by the secreted urease generating ammonia but the pathogen has to quickly migrate into the mucous layer of the stomach for long-term survival [116].

#### 2.10.2. Vaccines

Initial vaccination strategies were based on antigens derived from secreted proteins like urease [117,118], VacA [119], CagA [120], and catalase [121], that reduced bacterial load upon infection with *H. pylori*. However, there was room for improvement and the review will discuss 3 approaches.

In the first approach based on type IV secretion system antigens, Chehelgerdi and Doosti [122] used CagW protein as a probable functional analog of the structural VirB6 protein from the type IV secretion system. In the plasmid-based DNA vaccine strategy, the gene coding CagW protein was delivered in pcDNA-3.1 at 100 µg/mouse of plasmid encapsulated or not in chitosan nanoparticles into mice intramuscularly 5 times within 45 days. Challenge was with 1 × 10^9^ CFU of *H. pylori* via p.o. route. Remarkably, the bacterial load in the vaccinated animals was reduced by 8 logs in the liver and gastric tract as determined by bacterial colony counting. Encapsulation of the plasmid improved bacterial load reduction by at least a log as compared to the non-encapsulated plasmid immunization.

In the second strategy, a flagellar protein fused with urease was used. Several epitopes of flagellin with *H. pylori* urease were fused to cholera toxin subunit B [123] and used to immunize BALB/c mice. The constructs used were cholera-toxin B-urease fusion (200 μg), and the full cholera toxin B-urease-chimeric flagellin fusion (50–250 μg protein) 4 times at 1-week intervals. The mice were challenged with 1 × 10^9^ CFU of *H. pylori* 26695 strain orally 7 times over 2 weeks. The full fusion constructs given at the highest dose reduced gastric load of *H. pylori* by 2 orders of magnitude as judged by PCR. However, histopathological examination of the tissue showed that the pathogen was able to exert damage in the gastric tract despite vaccination.

In the third strategy, the approach used nanoparticles. The antigen described previously was encapsulated in acid-resistant nanoparticles based on D,L-lactate-co-glycolic acid polymer and used to immunize BALB/c mice at 100 µg protein/animal 4 times with 1-week intervals via the oral or intraperitoneal routes [124]. Two weeks after the final immunization, the animals were challenged orally with 0.4 × 10^9^ CFU of *H. pylori* SS1 strain via the oral route. The best results were obtained for the oral route of immunization with a certain type of nanoparticle formulation capable of reducing bacterial load in the gastric tissue by 3 logs as compared to the non-immunized animals. The i.p. route did not show any statistically valid bacterial load reduction in animals. As before, the histopathological examination of the gastric tissue revealed that the vaccination did not offer complete protection from *H. pylori* infection.

### 2.11. Legionella pneumophila

The bacterium is a Gram-negative aerobic facultative intracellular pathogen of humans and amoebae. It is also an etiological agent of Legionnaires’ disease of humans. There is no commercial vaccine available and the therapy relies on antibiotics.

#### Vaccines

The development of vaccines including antigens from the icm/Dot type IV secretion system had not been reported in the literature. Instead, the strategies used different elements of the pathogen and vaccine designs are shown for 2 representative studies based on known virulence factors.

In the first one, 2 recognized virulence factors, the major outer membrane protein encoded by the *ip* gene [125,126] and type IV pilin E [127], were used separately and as a fusion in a mouse A/J model of infection [128]. The animals were injected i.m. with 50 μg of pCDNA3.1 plasmid with cloned separate genes or their fusion and challenged with 2 × 10^7^ CFU of *L. pneumophila*. The fusion construct could protect all animals from death for at least 10 days while the separate proteins protected 30–40% of the animals. The fusion could be a good candidate for *L. pneumophila* vaccine design.

In the second strategy, the inclusion of peptidoglycan-associated lipoprotein (PAL) specific for *L. pneumophila*, type IV pilin E (PilE), and a key component of flagella FlaA into a plasmid-based DNA vaccine [129] as a fusion PAL-PilE-FlaA was tested in a BALB/c model of infection. The genes coding for the fusion were cloned into the pcDNA3.1 plasmid and animals were immunized with 50 μg of the plasmid/animal 3 times with 2-week intervals. Challenge was performed with 2 × 10^7^ CFU of *L. pneumophila* serogroup 1 (American Type Culture Collection, Manassas, WV, USA; no. 35133) via the i.v. route. The fusion construct could completely protect mice from the lethal dose of the bacterium for 10 days. Histopathological examination of lung tissue showed that there was an injury to the lung tissue upon challenge despite the vaccination.

### 2.12. Mycobacterium tuberculosis

The bacterium is an aerobic human pathogen and the aetiological agent of tuberculosis. The disease is fatal if left untreated with antibiotics. It is estimated that about a third of the human population is infected with the bacterium (https://www.who.int/teams/global-tuberculosis-programme/tb-reports). There is an FDA-licensed human live vaccine based on the attenuated bovine strain of *M. bovis* BCG [130]. However, some countries do not carry out vaccination with this strain [131].

#### 2.12.1. Secretory Systems

The pathogen has a general, Tat, and the type VII (EXS) secretion systems. The last one is critical for its virulence and deletions in it are found in the attenuated *M. bovis* BCG strain [132,133,134]. *M. tuberculosis* has 5 EXS systems [133] of which the first one, EXS-1, is the most important for its virulence [135,136], in addition to the EXS-2 and EXS-3 systems [132]. The EXS-1 system uses EccA1 ATPase to drive the secretion across bacterial membranes of, among other effectors, ExsA1 with help of ExsB1 as a probable chaperone. The secretion also requires EspA1/EspC1 complex presence. The EccA1 is critical for secretion and its deletion abolishes the transport [137], leading to attenuation of virulence.

Analysis of bacterial secretions showed that ExsA1 is critical for *M. tuberculosis* virulence and it contributes to mammalian-like membranes lysis activity under low pH conditions [138]. The phagosome lysis is contact-dependent and requires phthiocerol dimycocerosates [139,140].

It has been shown that *M. tuberculosis* infection results in the secretion of large (up to 20% of total secreted protein) amounts of antigen 85 B (Ag85B) [141]. The protein was shown to be a good immune target for the host [142].

#### 2.12.2. Vaccines

Recombinant protein-based *M. tuberculosis* vaccines based on the ESAT-6 and Ag85 antigens were used in many vaccine constructs [143,144,145,146,147,148]. However, a candidate better than the BCG live vaccine has been demonstrated only once. A fusion with the attenuated latency-associated protein Rv2660c (fusion Ag85B-ESAT6-Rv2660c) was used as a vaccine H56 [149]. The CB6F1 mice were immunized 3 times with the fusion in doses 0.01–10 μg/animal and challenged intranasally with the *M. tuberculosis* Erdman strain after 6 weeks. Bacterial load in lungs was found to be the smallest for the 5 μg dose and was about a log smaller than for the BCG vaccine. Boosting the BCG vaccination twice with the H56 fusion construct 3 months after the vaccination showed a log-reduced bacterial load in mice lungs 12 weeks post-boost. The system could be used pre- and post-exposure but the latter offered only a 1–2 log reduction of bacterial load in the lungs. The H56 recombinant vaccine could be a potential additional candidate for a hybrid vaccination strategy.

### 2.13. Proteus mirabilis

The bacterium is a Gram-negative facultative anaerobic pathogen infecting predominantly humans. It is frequently associated with urinary tract infections connected to catheters. There is no commercial vaccine available and the only therapy is with antibiotics.

The pathogen secretes urease to help it survive in the urinary tract [150]. The released ammonia is a source of nitrogen for bacteria and raises pH that in turn leads to the formation of stones inside the urinary tract or on catheters [151,152,153,154,155,156].

Infection of bladder epithelial cells by *P. mirabilis* requires specialized fimbriae, the uroepithelial cellular adhesion (UCA), one of 17 types possessed by the pathogen [157], and AipA protein, an autotransporter, for internalization [158,159]. Potential lysis requires secreted hemolysin HmpA, activated and transported with the help of HmpB [160,161], and a toxic agglutinin Pta [159].

The pathogen can establish infection 30 min post-inoculation in a mouse model [162] and cell culture [163]. The speed is essential for initial escape from infiltrating neutrophils from which it is also later protected by forming extracellular clusters in the urinary tract [162]. The formation requires urease activity and mannose-resistant *Proteus*-like (MR/P) fimbriae [162].

#### 2.13.1. Secretory Systems

Analysis of *P. mirabilis* genomes showed that the bacterium has type I, II, III, V, and VI secretion systems.

*P. mirabilis* secretes proteases with gelatinolytic activity [164]. However, the specific secretion system responsible for their secretion has not been identified. It is known, however, that the ZapA protease is essential for infection in a mouse model as deletion of the gene resulted in a significant reduction of bacterial loads in a mouse model of UTI [165] and a decrease in histopathology of prostate tissue in a rat model of prostatitis [166].

The type III secretion system present in the *P. mirabilis* HI4320 strain is expressed in this strain [167]. However, deletion of the essential ATPase or a negative regulator did not influence virulence in the ascending urinary tract infection (UTI) [167]. Therefore, the system is likely not important in the overall virulence of the pathogen.

The *P. mirabilis* HI4320 strain has 3 subclasses of type V secretion systems: Va, Vb, and Vc. The Va, a classical autotransporter system, likely secretes the toxic Proteus agglutinin Pta [157]. The hemolysin HpmA and its partner HpmB are secreted most likely by the two-partner secretion system type Vb [157]. The type Vc is responsible for the secretion of trimeric autotransporters, of which the *P. mirabilis* HI4320 has three [168]: the “adhesin-like” AipA, the “agglutinating adhesin-like” TaaP, and another one.

The type VI system is used to kill other bacteria, including other strains of *P. mirabilis*. Transposon mutagenesis of *P. mirabilis* HI4320 strain showed that the type VI secretion system has no role in virulence in a mouse model of ascending urinary tract infection (UTI) [169].

#### 2.13.2. Vaccines

Early attempts to develop a vaccine relied upon a continuous passaging of the pathogen through animals and using the attenuated strain as a vaccine again the starting parental strain. The strategy was used successfully by Jones via the oral route [170]. Later strategies used recombinant proteins, alone or fused with cholera toxin B (CTB) subunit or expressed on the surface of an attenuated different pathogen.

The OmpA outer surface protein was used by Zhang et al. [171] and Cui et al. [172]. In the work by Zhang et al. [173], natural adjuvant Taishan *Pinus massoniana* pollen polysaccharide (TPPPS) was used together with recombinant OmpA to immunize chicken. The SPF chickens were immunized via the s.c. route with increasing amounts of OmpA (50–150 μg) in adjuvant 3 times within 9 days and challenged orally with 100 LD50 of *P. mirabilis* Q1 strain a week after the last booster. The mortality of vaccinated chicken was reduced to 10% with the use of TPPPS adjuvant as opposed to around 60% when using the protein alone.

The protective efficacy of natural OmpA in protection against *P. mirabilis* infection was investigated by Cui et al.’s group [172]. The immunization was performed as described previously but the amount of protein was increased (0.5–2 mg) and the challenge was as described previously [171]. The inclusion of TPPPS adjuvant reduced the mortality rate to below 10% as opposed to under 30% when using the OmpA protein alone.

Vaccines using mannose-resistant fimbriae (MR/P) fusion with type I fimbriae from uropathogenic *E. coli* (UPEC) were used by Habibi et al. [174]. The group used monolauryl phosphate (MPL) adjuvant with 25 μg of protein total/animal injected via nasal route and boosted twice with a week’s interval into BALB/c mice. The immunization could reduce bacterial load in a mouse bladder by 5 orders of magnitude while the unfused combination reduced the load by only 3 orders.

A similar reduction was observed when flagellar FliC and *P. mirabilis* MrpA from the mannose-resistant fimbriae (MR/P) were used [175]. BALB/c mice were immunized with 30 μg protein/animal intranasally and challenged transurethrally into the bladder with 1 × 10^8^ CFU of *P. mirabilis* HI4320 strain 1 week past the immunization. Combination of proteins, addition or fusion, could reduce bacterial load in the bladder by 4 logs while separate proteins could reduce the load in the bladder by 2–3 logs only. Reduction in the kidney was 3 logs for the combination and 2 logs for the separate proteins.

Habibi et al. [176] used FimH from UPEC and MrpH from MR/P fimbriae from *P. mirabilis* to evaluate as potential vaccine candidates against uropathogenic infection (UPI). BALB/c animals were immunized with 25 μg/animal of proteins and their combinations via the s.c. route twice with 2-week intervals and challenged transurethrally with 1 × 10^8^ CFU of *P. mirabilis* or UPEC two (UPEC) or seven (*P. mirabilis*) days post-inoculation. Immunization with the fusion protein could reduce bladder bacterial load by 5 logs in the bladder and 4 in the kidney for the UPEC, and 6 and 5 in the bladder and kidney, respectively, for the *P. mirabilis* infection.

Change or immunization route to intranasal for the proteins gave worse effects than for the transurethral route as reported by Li et al. [177]. CBA mice were immunized intranasally, transurethrally, or orally with 1 × 10^9^ CFU formalin-killed *P. mirabilis* HI4320 strain or 100–200 μg MR/P fimbriae coupled to cholera toxin with complete Freund’s adjuvant and boosted twice with week intervals using incomplete Freund’s adjuvant. Mice were challenged on day 21 with 5 × 10^7^ CFU *P. mirabilis* HI4320 strain via the transurethral route. The intranasal immunization with the major subunit MrpH of MR/P fimbriae was mildly effective (2 logs) in reducing bladder load and much better at reduction of the kidney load (4 orders) of *P. mirabilis* in a mouse model of infection.

However, the fusion of MrpH with cholera toxin chimeric fusion of A2 and B subunits was much more effective in reducing the bacterial load (4.5+ for bladder and 3+ for kidney) via the intranasal route in the same model [178]. CBA mice were immunized as described above using the cholera toxin chimera A2 and B subunits as a mucosal adjuvant and challenged identically.

In another study on *P. mirabilis* vaccines, Bameri et al. [179] used a fusion of MrpH (*P. mirabilis*) and FliC (*E. coli*). BALB/c mice were injected s.c. with 25 µg of proteins or their fusion and boosted twice every 2 weeks. Animals were challenged 1-week post-booster with 1 × 10^7^–10^8^ CFU of *P. mirabilis* transurethrally. The fusion protein was very effective in reducing the bacterial count of *P. mirabilis* in the bladder (5 orders) and kidney (3 orders).

Many studies routinely included flagellin components as adjuvants acting on TLR receptors. However, Scavone et al. [180] showed that such inclusion decreased the protective effect of *P. mirabilis* MrpA subunit of MR/P fimbriae. CD-1 mice were immunized intranasally with 30 μg (MrpA) or 7 μg (flagellin) and their combination per animal and boosted 4 times over 24 days. One week after the last booster, the animals were challenged with 1 × 10^8^ CFU *P. mirabilis* Pr2921 transurethrally via a catheter.

The cancellation effect was reported only for the MrpA-FliC fusion but not for other fusions reported earlier. Therefore, the result may be connected to the particular combination of these two proteins.

A live vaccine approach was used by Scavone et al. [181]. The group showed that a live vaccine based on *Salmonella* Typhimurium expressing *P. mirabilis* MrpA subunit fused to tetanus toxin TetC was effective. BALB/c mice were immunized intranasally with 1 × 10^9^ CFU of *S.* Typhimurium expressing the fusion protein and boosted after 28 days. Mice were challenged with 2 × 10^8^ CFU of *P. mirabilis* Pr2921 transurethrally 28 days past the booster. The live vaccine could reduce colonization of the urinary tract in mice by at least 3 orders of magnitude in kidneys and the bladder when using the intranasal vaccination route.

A different strategy designed to select B- and T-cell epitopes by the immune system was used by Choubini et al. [182]. The group used MrpA, uroepithelial cell adhesin (UcaA), and Pta fragments from *P. mirabilis* selected for optimal B- and T-cell epitopes. BALB/c mice were immunized with 50 μg of total protein, separate or combinations, mixed with AdaVax adjuvant, boosted twice every 2 weeks, and challenged with 1 × 10^8^ CFU/mL of *P. mirabilis* HI4320 strain 1 week after the last booster. The squalene-like adjuvant AddaVax was used to elicit Th1- and Th2-type immune responses. Immunization with the PtA-MrpA-UcaA fusion protein of all fragments was able to reduce the bacterial load of *P. mirabilis* by 6 orders in the bladder and 5 in kidneys in a UTI model of infection. It is the most effective vaccine construct against *P. mirabilis* reported in the literature to date.

### 2.14. Pseudomonas aeruginosa

The bacterium is a Gram-negative opportunistic pathogen infecting humans and animals. It is a part of human commensal flora but in cases of weakened immunity, the pathogen can be deadly if left uncontrolled. The situation is specifically problematic in the hospital settings in Intensive Care Units (ICUs) due to its broad drug resistance and ease of transmission. The bacterium is on the list of ESKAPE pathogens designated as a public health problem. An approved therapy relies on antibiotics. Indirect approaches, including virulence blockers, have not been available commercially despite excellent work by many groups [183,184,185,186,187,188,189,190,191].

#### 2.14.1. Virulence Systems

The pathogen has multiple virulence systems according to the KEGG database (https://www.genome.jp/kegg-bin/show_pathway?pae03070): type I, II pathogenic and general, type III, Tat and type VI. Type II and III are targeting mostly host cells while type VI- bacterial flora, including that of the host [192]. *P. aeruginosa* has 3 types of type VI secretion systems [193] and their roles in infection are beginning to emerge [194]. The most studied system is encoded by the H1 locus [195,196].

Overall architecture and function of the type III secretion system in the pathogen have been described in detail [23,197,198] and type II has been detailed in *P. aeruginosa* as well [199,200,201,202,203]. In an animal model of infection, the type III system is the most important one for virulence with type II providing a minor role [204]. The first system secretes 4 effectors: bifunctional ExoS and ExoT enzymes with GTPase-activating protein and ADP-ribosyltransferase activities [205,206], ExoY, an adenylate cyclase [207], and ExoU, a potent cytotoxin with a phospholipase A2 activity [208,209]. The ExoU effector is the most important for virulence in a mouse model of infection [210]. The second system secretes, among others, guanylate cyclase ExoA, proteases lasA/B, and many other factors [211].

In some clinical strains lacking a type III secretion system, the pathogen uses a two-partner secretion (TPS) system to deliver a pore-forming exolysin (ExlA) [212].

The fimbriae have been also recognized as virulence factors [213,214,215]. Accordingly, they have been part of vaccine designs for other pathogens described previously and *Pseudomonas aeruginosa*.

The pathogen is known to express 3 kinds of type IV pili. The system is very much conserved and present in many bacteria. In *P. aeruginosa*, the system provides flagellum-independent motility [216,217].

#### 2.14.2. Vaccines

There is no commercial available despite many years of research. Published vaccine strategies relied on different principles, ranging from inactivated whole cells to single recombinant proteins coupled with a carrier or alone.

##### Flagellins

*P. aeruginosa* expresses 2 types of flagellins: type a and type b based on the serotype [218]. Type b is conserved among pathogens while type a is very variable except for one epitope a0 [219]. Behrouz et al. [220] used bivalent flagellin composed of both types to immunize mice in a pneumonia model of infection. The BALB/c mice were immunized intranasally with 2 μg of total flagellin at weekly intervals and challenged with 2 × 10^7^ CFU of *P. aeruginosa* PAK and PAO1 strains via the same route. The immune response was predominantly through CD4^+^ T lymphocytes and the strategy could protect animals against the fatal dose of pathogen up to 90% when immunized with both flagellins. The immunization with recombinant flagellins would increase survival rates up to 90% while reducing liver and spleen bacterial loads by 3 orders of magnitude. However, blood loads were reduced only by 2 orders while skin bacterial loads were unaffected.

Hegerle et al. [221] coupled recombinant flagellins from *P. aeruginosa* with a purified outer polysaccharide (OPS) and used it for immunization of mice in a wound infection model. The Crl:CD-1 mice were injected i.m. into the right gastrocnemius with 2.5 μg or 5 μg of OPS conjugate 3 times over 28 days. Passive protection with post-immunization antibodies would increase the survival rate up to 50% for animals challenged via the i.p. route with 2.5–10 LD50 *P. aeruginosa* strains MA, PA, and PAO1.

Inclusion of type IV pilin into a vaccination strategy with type a and type b flagellins was used by Hashemi et al. [222] in a burn wound model. The animals were immunized via the s.c. route with proteins, 5 μg/protein, 3 times over 28 days and challenged s.c. under the created burn area on day 42 with 3–5 × 10^2^ CFU of *P. aeruginosa* PAK or PAO1 strains. At least 90% of the animals survived the challenge with either strain when immunized with the trivalent vaccine. Removal of one component could decrease survival rates to 50%, depending on the component and the strain.

Using the same burn model, Korpi et al. [223] tested a combination of flagellin b and pilin A against clinical isolate. The BALB/c mice were immunized with 10 μg of each protein/animal and challenged with 3–5 × 10^2^ CFU of *P. aeruginosa* PAO1 and clinical isolate strains as described previously [222]. The use of only recombinant flagellin b could protect at least 70% of animals while immunization with the pilin A protein would drop protection to 50% against the clinical isolate.

##### Porins

The proteins are outer surface antigens present on the surface of bacteria. Their function is frequently a transport of other small molecules, including β-lactam antibiotics [224]. An analog of OmpA, the PA0883 protein identified by reverse vaccinology, was used as a vaccine candidate in an animal model of pneumonia [225]. BALB/c mice were immunized i.m. with 30 μg protein/animal 3 times over 28 days and were challenged on day 35 with 5.0 × 10^6^ CFUs of PAO1 intratracheally (pneumonia model) or 3.5 × 10^7^ CFUs of PAO1 intravenously (sepsis model). Immunization with the recombinant protein could protect up to 70% of animals from death in the sepsis model and 50% in the pneumonia model.

Yu et al. [226] used a fusion of Outer membrane protein F (OmpF) with a VP22 peptide in a DNA vaccine to facilitate the cellular spreading of antigen after immunization [227]. The BALB/c mice were immunized via the i.m. route with 20 μg of plasmid and boosted 2 times with 2-week intervals and then challenged with 5 × 10^6^ CFUs of *P. aeruginosa* PAO1 strain. The immunized mice had about a 50% survival rate after immunization with the C-terminal fusion of VP22 and none for the unfused and N-terminally-fused constructs after 10 days.

The strategy to use the fusion of OprF with OprI fragments from *P. aeruginosa* was tested in a human trial of non-surgical ICU patients [228]. Patients were immunized via the i.m. route with 100 μg of protein twice with a 1-week interval. Unfortunately, the trial showed no clinical benefit of the vaccination in protection against *P. aeruginosa*-induced mortality. The data is in strong contrast with the work of Hasan et al. [229] that showed significant protection of a mixture of OprF/OprI in a mouse model of infection.

Zhang et al. [230] also tested a fusion of OprF/OprI on the surface of attenuated *Salmonella* Typhimurium. BALB/c mice were immunized with 2 × 10^10^ CFU of recombinant *S.* Typhimurium orally or 2 × 10^8^ CFU subcutaneously and boosted with the same doses after 2 weeks and then challenged with 5 × 10^7^ CFUs of *P. aeruginosa* ZHDL9 virulent strain. The immunization could provide up to 80% survival rate in a mouse model of intranasal *P. aeruginosa* infection for the s.c.-immunized animals and only 40% for the orally immunized mice over 15 days.

In another use of the OprF/OprI construct, fusion to a heptameric hemolysin (Hla) from *S. aureus* was used to immunize BALB/c mice (40 μg/animal, i.m. route) and boosted twice within 21 days. One week after the last booster, mice were challenged intratracheally with 1 × 10^7^ CFU of *P. aeruginosa* PAO1 strain. The oligomerized fusion construct increased the survival rate of animals from 30% to 80% in a mouse model of pneumonia *P. aeruginosa* infection when compared to the protein alone [231]. In previously described human trials for the protein alone, immunization with the protein alone did not offer any improvement in mortality rates for the ICU patients [228].

To improve the survival rate for the OprF/OprI fusion protein, a PcrV fragment was added [232]. The OprF-OprI-PcrV fusion and separate components were injected via the s.c. route into BALB/c mice (10 μg/animal) and boosted twice over 28 days. The animals were challenged via the s.c. route with 2–10 LD50 of *P. aeruginosa* strains PAO1, PAK, and R5 using a burnt wound infection model. At least 75% of animals were protected up to 10 LD50 challenge doses against all strains for 10 days when using the complete fusion protein. Bacterial loads were reduced by at least 4 logs when examining skin and liver or spleen and kidney of the animals immunized with the full fusion protein.

To find peptides from *P. aeruginosa* stimulating Th-17 immune response, Wu et al. [233] used a small peptide library to identify OprL as stimulating IL-17 secretion upon immunization. The mutated version, reOprL, was designed and used as a vaccine candidate. Testing was performed in a mouse model of *P. aeruginosa* infection [234]. The C57BL/6 mice were immunized intranasally with 50 μg of protein and curdlan combinations and boosted 2 times over 28 days. The animals were challenged intratracheally with *P. aeruginosa* strains PA XN-1, PA01, PAK, PA27313, and PA27315 at 1.0 × 10^7^, 5.0 × 10^7^, 2.0 × 10^7^, 2.0 × 10^7^, and 2.5 × 10^7^ CFU, respectively. As predicted, the Th-17 response was independent of the tested pathogen strain but the survival rates after vaccination were at best 60% 7 days post-challenge.

The use of another porin, OprH, in a mouse model of *P. aeruginosa* infection was also tested. BALB/c mice were immunized intranasally 3 times at weekly intervals with 10 μg of refolded OprH protein and curdlan as an adjuvant and later challenged with PA14 (2 × 10^7^ CFU per mouse) or PA103 (2 × 10^6^ CFU per mouse) virulent strains. Immunization with a refolded OprH protein with curdlan could protect up to 50% of the animals in the best case [235].

##### Type III Secretion System Effectors

The strategies are based around the tip protein, PcrV, as it is essential for effector delivery and is typically exposed to antibodies on the pathogen’s surface. Immunization with the rPcrV and then challenge with a fully virulent strain administered with the antiPcrV antibodies could completely protect the mice from a lethal dose of *P. aeruginosa* in a lung model of infection [236]. Removal of the preimmunization with PcrV or change of the preimmunization antigen to ExoU or PopD decreased the survival rates substantially. The data agree with other studies where administration of PcrV during vaccination was not sufficient to protect the animals completely based on the survival rates [237,238,239,240,241,242].

To help remedy the situation, a chimeric construct based on PopB/PcrH proteins mixed with curdlan and encapsulated in poly-lactic-co-glycolic acid (PLGA) to control delivery of vaccine candidate was tested in mice [243]. The FVB/N mice were immunized intranasally with 20 μg of PopB/PcrH and their fusion per animal weekly for 3 weeks and challenged intranasally with 1 × 10^6^ CFU of *P. aeruginosa* strain ExoU+ PAO1 three weeks after the last immunization. Encapsulation of antigens in PGLA improved protection from death from 50% to 70% when compared to the nonencapsulated antigen for at least 6 days. Bacterial burden in lungs was also reduced by 5 logs for the encapsulated version. However, the IL-17 secretion stimulation upon vaccination was inferior to the stimulation observed for curdlan.

The translocator proteins as vaccine antigens were used also by Das et al. [244]. The group used a self adjuvating construct composed of a fusion of PcrV-PopB, LT1-PcrV-PopB, or separate proteins with a separate addition of mutated heat-labile enterotoxin (dmLT). BALB/c mice were immunized intranasally with different amounts of proteins (1–20 μg/animal) 3 times over 28 days and challenged 28 days after the last immunization with 2.5 × 10^7^ CFU/30 µL of the PA15808959 strain. The exact challenge dose per animal was not given by the authors. The immunized animals showed at least a 5-log reduction in bacterial load in the lungs and a strong IgA and IgG induction. The load reduction of bacteria in the lungs for the animals immunized with A subunit of the heat-labile enterotoxin fused to LcrV and PopBV was 6 logs when the immunization was performed with 20 μg of the protein. The mortality rates were not listed in the work.

Search for the most immunogenic proteins was performed by Xu et al. [245]. The group used a random screen of genomic library constructs of *P. aeruginosa* strains isolated from convalescent patients fused with maltose-binding protein (MBP) to select the most immunogenic proteins. Not surprisingly, the type III secretion system needle protein PscF was the only protein from the type III secretion system selected for the group of 13 proteins and later tested for protection against the fully virulent *P. aeruginosa* strain in a mouse model of lung infection. Purified MBP fusions were injected into BALB/c mice at 50 μg/animal and boosted twice over 21 days. Animals were challenged intranasally with 1 × 10^9^ CFU of *P. aeruginosa* XN-1 strain 10 days after the last immunization. The PscF-based construct was the best vaccine candidate and protected 90% of the animals from the pathogen based on the survival ratio for 6 days.

##### Pilins

Investigation into the role of pilin A (pilA) in immune protection against *P. aeruginosa* was reported by Banadkoki et al. [246]. BALB/c mice were immunized with recombinant pilin A (6 μg/animal) with or without naloxone (6 μg/g body weight) subcutaneously 3 times over 28 days and challenged intranasally with 3–5 × 10^7^ CFU of *P. aeruginosa* two weeks after the last booster. Immunization with pilA protected at least 70% of animals from death in an intranasal model of *P. aeruginosa* PAO1 strain infection in mice when used with naloxone and only 40% when the opioid was not used. The same result was seen for a clinical isolate of *P. aeruginosa*.

##### O-Type Antigen

The approach used an engineered strain of *Salmonella* Typhimurium *ΔwecA* mutant [247]. The strain expressed the heterologous O-antigen from *P. aeruginosa*. BALB/c mice were immunized orally with 1.3 × 10^6^ CFU or intraperitoneally with 8 × 10^2^ CFU of mutant *Salmonella* and challenged intranasally with 2.6 × 10^5^ to 3.4 × 10^6^ CFU of *P. aeruginosa* PA103 strain. For the orally-immunized animals, none survived the challenge but for the i.p.-immunized animals, the survival ranged from 20% to 90% for the lower doses. An increase of the challenge dose to 6.0 × 10^5^ CFU showed that none of the animals survived the challenge.

##### Outer Membrane Vesicles

The vesicles are normally released by *P. aeruginosa* and other pathogens. They are composed of lipopolysaccharide and other outer membrane proteins while the inside contains nucleic acids and many cytoplasmic proteins [248,249]. Zhang et al. [250] used *P. aeruginosa* vesicles to immunize mice and later challenge with the original or other *P. aeruginosa* strains in a mouse model of acute lung infection. BALB/c mice were immunized via the i.m. route with 30 μg of OMVs per animal and boosted twice within 21 days. Challenge was performed intratracheally with 5.0 × 10^6^ CFUs of PAO1 strain one week after the last booster. The animals were protected from the lethal dose of the PAO1 strain up to 90% based on the survival rates after 10 days. However, protection against 3 clinical strains: XN-01, BJ-16, and KM-9, ranged from 60% to 90%.

##### Auxotrophs

Strategies based on auxotrophic vaccines have been known for a long time. However, there have not been many reported with a high success ratio. In the approach by Cabral et al. [251], the genes responsible for conversion of L-glutamate into D-glutamate, *murI*, and the potential rescue gene *dat* coding for D-alanine aminotransferase, have been removed. The resulting strain cannot make a full peptidoglycan structure due to the lack of D-glutamate. The mutations were stable and the strain was used as a live vaccine against fully virulent *P. aeruginosa* strains in a mouse model of infection. BALB/c or C57BL/6 mice were immunized intravenously with 2 × 10^7^ CFU of the mutant strain and boosted after 2 weeks. The animals were challenged with 2 × 10^7^ CFU of PAO1 strain on day 25, 4 × 10^7^ CFU of the same strain on day 28 (C57BL/6 strain), or 4 × 10^7^ CFU of PA28562 strain on day 25 (C57BL/6 mice). The vaccination almost completely protected the animals from death and reduced bacterial loads in the liver, spleen, and lungs by at least 4 logs. To improve protection, an extra booster on day 28 and a challenge on day 35 were used. In that case, all animals were protected for at least 7 days. In the pneumonic model of infection with *P. aeruginosa* in mice, the strategy based on a single gene deletion, *murI*, was only partially successful after different immunizations [252]. BALB/c mice were immunized with 2 × 10^8^ CFU of mutant and boosted on days 8 and 15. Challenge was performed on day 42 with different strains and amounts using the intranasal route. The PAO1 strain was fully protected against for 7 days when the 5 × 10^5^ CFU challenge was used but only 20% when the 4–10^7^ CFU of the pathogen was applied. Reduction of booster shots to one and application at day 14 had mixed effects on different PA strains. Challenge with PAO1 ExoU+ (7 × 10^5^ CFU), PA14 (1 × 10^6^ CFU), ST235 (3 × 10^7^ CFU), and PAO1 (2 × 10^6^ CFU) offered at least 90% protection over 7 days except for the ST235 strain where the immunization was protective for only 30% of the animals.

### 2.15. Salmonella enterica

The bacterium is a facultative anaerobic Gram-negative pathogen, frequently infecting humans. It is frequently associated with food poisoning, specifically raw eggs [253,254]. In developing countries, the pathogen is an etiological agent of enteric fevers, caused predominantly by 2 serovars, Typhi and Paratyphi. There are 3 vaccines recommended for Typhi serovar: (i) typhoid conjugate vaccine (TCV); (ii) unconjugated Vi polysaccharide (ViPS) vaccine, and (iii) live attenuated Ty21a vaccine, but no recommendation exists for the Paratyphi serovar [255].

Recognized virulence factors include effectors secreted by the virulence systems, fimbriae, and the capsular polysaccharide Vi. Serovar Typhi also encodes a cytolethal distending toxin Cdt [256]. The toxin is composed of 2 subunits, A and B.

#### 2.15.1. Virulence Systems

The bacterium has type I, general, type III, Tat, and type VI virulence systems (https://www.genome.jp/kegg-bin/show_pathway?sty03070). There are 2 main types of III secretion systems coded on separate *Salmonella* Pathogenicity Islands (SPI): SPI-1 [257] and SPI-2 [258]. Both systems cooperate in killing macrophages and subsequent escape of pathogen from them. However, the first one is responsible for the cellular invasion of host epithelial cells while the second one- for the establishment of intracellular replication [259,260,261]. Type VI is necessary for the establishment of colonization in the intestines [262] and cytotoxicity to human epithelial cells [263].

#### 2.15.2. Vaccines

##### Cytolethal Distending Toxin B

There is only 1 report on the use of Cdt as a vaccine candidate. Thakur et al. [264] used the B subunit of Cdt to immunize mice against *S. enterica*
*S.* Typhi. BALB/c mice were immunized via the i.p. route using 50 μg protein/animal incomplete Freund’s adjuvant and boosted with 25 μg of protein in Freund’s incomplete adjuvant 3 times within a month. The animals were challenged with 10^9^ CFU of *S.* Typhi per animal via the same route. Surprisingly, 80% of animals survived the challenge for 30 days and histopathological examination showed tissue restoration within the liver and spleen of the immunized animals after 7 days post-challenge.

##### Secreted Effectors

There are 5 reported cases of using T3SS effectors as vaccine candidates in this review. In the first one, Hu et al. [265] used a fusion of *S. enterica* T3SS-2 effector SspH2 (E3 ubiquitin ligase) with *E. coli* EscI protein as a live *Salmonella* vaccine candidate in a mouse model of i.v. infection with *S. enterica*. C57BL/6 mice were immunized via the i.v. route with 10^7^ CFU of engineered *Salmonella* and challenged with 1 × 10^5^ CFU via the i.p. route with *S. enterica* D6 fully virulent strain 1-week post-immunization. Surprisingly, the addition of the EscI protein enhanced the survival rate to 100% vs. 40% for the construct lacking the EscI part. Bacterial loads in the spleen and liver were only 1.5 logs reduced upon vaccination with the SspH2-EscI construct while no statistical difference was seen for the SspH2 construct alone.

In the second case, Martinez-Becerra et al. [266] used effectors from T3SS-1 and T3SS-2 to test as vaccine candidates in a mouse model of *Salmonella* Enteritidis infection. They used tip and first translocator fusions from T3SS-1 (SipD-SipB) or T3SS-2 (SseB-SseC) and their combination in an oral gavage model. BALB/c mice were immunized with recombinant proteins (20 μg/animal) via the i.m. route and boosted twice within 28 days. Challenge was performed via the oral route with 2 × 10^8^ CFU of *S.* Typhimurium SL1344 or 5 × 10^7^ CFU of *S.* Enteritidis 125109 strains 28 days after the last booster. The combination formulation was the most protective and could increase the survival rate to 60% vs.10% for either construct alone for 20 days.

In the third case, Jneid et al. [267] used PrgI (needle) and SipD (tip) from T3SS-1 as antigens to immunize mice against *S.* Typhimurium in an oral gavage model of infection. BALB/c mice were inoculated via the s.c. or i.n. routes and boosted twice with 21-day intervals. In the s.c. route, the proteins (20 μg) were mixed with alum as adjuvant while in the intranasal route- 10 μg of protein was used with 1.5 μg of cholera toxin as an adjuvant. For the oral route, 300 μg of protein with 10 μg of cholera toxin was used and animals were boosted 2 or 3 times with 21-day intervals. Animals were challenged with 100 LD50 (~10^6^ CFU) of *S.* Typhimurium orally 42 days past the last booster. The best protection (60%) was observed for the oral immunization protocol with a triple booster for the SipD protein. Removal of the third booster decreased protection to 40% for the SipD antigen in the oral immunization protocol. A combination of both antigens could afford only 40% protection for the oral immunization protocol regardless of the extra booster. Intranasal immunization showed that the SipD antigen could offer 50% protection while the combination- only 40%.

In the fourth work, the IpaD (*Shigella*) and SipD (*Salmonella*) antigens were used in cross-protection from *Shigella flexneri* and *S.* Typhimurium [268]. BALB/c mice were immunized via oral route with 300 μg recombinant proteins mixed with 10 μg of cholera toxin and boosted twice with 21-day intervals or via the intranasal route with 10 μg of protein mixed with 1.5 μg of cholera toxin using the same schedule. The orally-immunized animals were challenged with 100 LD50 of virulent *S.* Typhimurium (~10^6^ CFU/mL) orally or with 100 LD50 of virulent *S. flexneri 2a* (~5 × 10^10^ CFU/mL) intranasally for the i.n.-immunized animals 42 days after the last booster. In the homologous protection from *Salmonella*, the SipD antigens offered 40–50% protection over 21 days while in the heterologous protection- the IpaD offered only 30% protection in the same period regardless of the immunization route. In the homologous protection from *Shigella*, the IpaD antigen delivered orally offered 60% protection while the intranasal delivery reduced the protection to 30%. In the heterologous protection from *Shigella*, the SipD antigen could protect 70% of the animals when delivered orally but only 50% when the route was changed to intranasal.

In the last reviewed case, Lee et al. [269] used a combination of SseB and flagellin to immunize mice against *Salmonella* via an oral route. C57BL/6 mice were immunized with a recombinant SseB (100 μg/animal), *Salmonella*-purified flagellin (100 μg protein +10 μg LPS), or their combination via i.v. route and boosted after 4 weeks. Challenge was performed with 1000 CFU of *Salmonella* SL1344 via the i.v. route or 5 × 10^7^ CFU *Salmonella* SL1344 orally 4 weeks after the last booster. Each protein did not protect the animals but a combination of both could protect the animal up to 20% based on survival rates for 100 days post-infection. Bacterial loads in the spleen and liver were reduced by 4 logs for the combination delivered intravenously but only by 1–2 logs when separate proteins were used. Change of the immunization route to oral reduced bacterial loads in spleen and liver by only 3 logs for the combination of antigens.

##### Invasin

To address the gap in vaccines against *S.* Paratyphi, a different protein antigen was used. The strategy was based on the fact that *S.* Typhi invasin is always present on the surface of bacteria, and its removal decreases the pathogen’s virulence [270,271]. In the mouse model of *S.* Paratyphi intragastric infection by Das et al. [272], BALB/c mice were immunized with 5 μg/animal of recombinant *Salmonella* Typhi Invasin (rSTIV) via the s.c. route and boosted 3 times every week. Challenge was performed with 5 × 10^6^ CFU of *S. Typhi* or 5 × 10^5^ CFU of *S.* Paratyphi via the oral route 8 days after the last booster. The rSTIV antigen offered 90% protection from the pathogen for 5 days, better than the 50% protection observed for the standard vaccine Vi-TT in the experiment. Bacterial load reduction in spleen and liver was less than 1 log, unfortunately. Challenge with *S.* Paratyphi showed that the rSTIV antigen could protect 70% of animals for 5 days. However, the challenge dose was equal to 10 LD50, about 10 times lower than the background cutoff for vaccines.

##### Porins

The proteins are Outer membrane protein F (OmpF) and OmpC present in *Salmonella* and, due to their conservation, frequently used as universal vaccine candidates for a given strain. The serovar Enteritidis is frequently present in poultry farms and a combination of those 2 proteins plus a total extract of outer membrane proteins (OMV) were used by Li et al. [273] as a candidate vaccine. Hyline White Chickens were immunized via the i.m. route with a total of 60 μg of a combination of rOmpF, rOmpC, and OMV with adjuvants with a booster after a week. Animals were challenged with 2 × 10^9^ CFU of *S.* Enteritidis LQSE171 strain via an unspecified route. The best combination included OMVs with recombinant OmpF which could fully protect the chicken from the pathogen as judged by survival rates after 14 days. The addition of rOmpC decreased the protection by about 10%. In general, the protection was at least 60% for the tested combinations. Bacterial loads of organs showed that the pathogen was mostly eliminated by day 14 and completely by day 21 when examining the spleen, ileum, and liver. Elimination from cecum was not possible, either.

In a variation of OMV-based antigens, encapsulation of OMVs with chitosan sulfate and the addition of flagellin was used as a vaccine candidate against *S.* Enteritidis. Chicken eggs were immunized with 1000 μg of chitosan-encapsulated nanoparticle vaccine [274] and flagellin by *in-ovo* vaccination. The animals were challenged at day 7 post-hatching with 1 × 10^9^ CFU/bird of *S.* Enteritidis via the oral route. The vaccine could completely protect the chicken from mortality but the bacterial load reduction in ceca was reduced only by a log.

##### Flagellins

Flagellins are frequently used as vaccine candidates as they are recognized by the innate immune system receptors. There are 2 examples described in the review.

Baliban et al. [275] conjugated the FliC flagellin protein to the core of normal O-deacetylated lipopolysaccharide from *S.* Enteritidis or the FliC deletion mutant and used it as a vaccine candidate. CD-1 mice were immunized with 2.5 μg of FliC or the conjugate via the i.m. route and challenged with 1 × 10^6^ CFU of *S.* Enteritidis via the i.p. route and boosted twice with 28-day intervals. Four weeks after the last booster, the animals were challenged with *S.* Typhimurium D65 strain 1–5 × 10^5^ CFU via the i.p. route. The normal LPS fraction fused with the FliC protein could protect at least 90% of mice in the challenge while the O-deacetylated LPS fraction could reach only 60% protection in the mouse model of infection. The results were obtained for the 5–10 LD50 challenge dose which is below the background 100 LD50 level encountered in vaccine studies.

An increase of immunization load with the FliC-LPS core fusion to 3× could protect 80% of mice from death when challenged with approximately 40 LD50 of the virulent strain [276]. CD-1 mice were immunized with 5 μg/animal of FliC alone or coupled with the core LPS and challenged with 1 × 10^6^ CFU of *S.* Enteritidis R11 strain via the i.p. route. When the alum or MPL adjuvant was used during immunization protocol, the survival rate was 80% for the adult mice but only 70% for the infants. An increase in the antigen dose to 3× during immunization and the use of MPL for the core LPS-FliC conjugate could boost protection to 90% for the adult-primed and 70% for the infant-primed animals. Removal of the *fliC* gene from the challenge strain also reduced animal protection as compared to the wt strain.

##### Live Pathogens

Strategies for creating live vaccines of *Salmonella* are varied. Classic deletion of amino acid biosynthesis genes surprisingly led to increased virulence in *Salmonella enterica* [277]. However, the aro- mutant has been used in chicken as a live vaccine (Vaxsafet ST^®^ in Australia) [278].

In China, the *S.* Paratyphi-based C500 live vaccine has been used to protect swine [279]. Its immune mechanism is not known, however.

Metabolic interference in essential pathways was tried by Datey et al. [280]. A replacement of folate biosynthesis pathway gene *folD* encoding methylenetetrahydrofolate dehydrogenasecyclohydrolase with a heterologous *fhs* gene encoding formyltetrahydrofolate synthetase in *trans* in *S.* Typhimurium strain was constructed. BALB/c or C57BL/6 mice were orally inoculated with 1 mL of 10^4^ CFU of TM*ΔfolD*/pACDH-*fhs* strain and boosted with 10^2^ CFU of the same strain after 7 days. Challenge was performed with 10^8^ CFU of STM-WT 7 days post booster. The immunization could protect mice against approximately 100 LD50 of the virulent strain in an oral model of mouse infection. The strategy could offer long-term protection but the clearance of bacteria from intestines was only 2–3 logs.

In another study of interference with essential metabolism, the pathway leading to cobalamine was altered [281]. Cobalamine biosynthesis-deficient mutants *ΔcobSΔcbiA* of *S.* Gallinarum were used in a chicken infection model. Brown-layer hens were inoculated with 10^8^ CFU of the mutant orally and challenged with 10^8^ CFU of the SG287/91 strain 20 days post-inoculation. The vaccine completely protected chicken from the fully virulent strain as judged by mortality rates. The protection was tested against 100 LD50 of the fully virulent strain, a background level in vaccine development.

Removal of part of the phosphotransferase system (*ΔptsI Δcrr*) necessary for carbohydrate transport from the medium into the cytoplasm of the pathogen was tested by Zhi et al. [282]. The protection was complete in the intraperitoneal (i.p.) and oral routes of immunization and could clear mice of the pathogen from mesenteric lymph nodes, spleen, caecum, and blood.

A search for virulence attenuation in *S.* Enteritidis was performed by transposon scan mutagenesis [283] and many targets were identified. Guo et al. [284] used an *ΔsptP* mutant of *S.* Enteritidis strain C50336 as a live vaccine candidate. SPF chickens were immunized with 10^6^–10^9^ CFU of the mutant via the i.m. route and challenged with 1 × 10^8^ CFU of the fully virulent Z-11 strain via the same route. Immunization with the mutant strain could protect against approximately 1000 LD50 of the fully virulent wild type pathogen. Bacteria were also cleared completely from the spleen and liver by day 14 post-infection in the vaccinated chicken.

Protein synthesis pathways interference was used in another attempt at live vaccine. Park et al. [285] used a *ΔyjeK* mutant deficient in the protein synthesis elongation step. BALB/c mice were injected with 10^4^ CFU/dose of the *S. enterica* Typhimurium 1120 strain via the i.p. route and challenged with ST2173 strain at 10^8^ CFU/dose (protection assay) or 10^10^ CFU/dose (survival assay) orally 28 days post-immunization. At the 100 LD50 challenge dose of the pathogen, all immunized animals survived. However, bacterial load reduction in the spleen was only 1 log as compared to the unvaccinated animals.

Direct interference with the virulence secretion system was tried for the type III secretion system by Yin et al. [286] where the full SPI-2 operon in *S.* Paratyphi A strain SPA017 was removed. BALB/c mice were inoculated with the SPA017ΔSPI2 mutant at 5 × 10^5^ CFU and challenged with 10^3^ CFU of the wt strain via the same route at unspecified time post-immunization. Immunization via the i.p. route of mice with the mutant could completely protect animals from at least 10^3^ CFU of fully virulent strain. Unfortunately, the challenge dose was only 10 LD50 which is below the background level for vaccine testing.

Double interference with aromatic amino acid biosynthesis and virulence system was tested for the double mutant *ΔaroC ΔSsaV* (SPI-2) of *S. enterica* Typhimurium. The construct was used as a potential live vaccine and progressed into Phase I clinical trials [287]. However, the mutant did not progress into commercial development due to poor protection.

Triple mutant targeting porin, carbohydrate metabolism, and type III secretion system were used by Li et al. [288]. The construct combined deletion from SPI-2 (*ΔspiC*) with OmpD porin (*ΔnmpC*) or rfaL involved in mannitol utilization (*ΔrfaL*) in a chicken model of *S.* Enteritidis infection. SPF Hyline White chickens were immunized with CZ14-1*ΔspiC∆nmpC*, CZ14-1*ΔspiC∆rfaL*. In a rare study, the animals were challenged with 10^5^ LD50 of virulent strain after vaccination with the combinations. At least 75% of chicken survived the challenge and the bacteria were eliminated from the cecum, ileum, spleen, and liver.

Another triple mutant as a live vaccine candidate was tried by Zhao et al. [289]. The group used a triple mutant *ΔfliC* (flagellin) *Δfnr* (regulator of fumarate and nitrite reduction) *ΔarcA* (regulator of metabolism in response to oxygen levels) as a live vaccine against *S.* Typhimurium in a mouse model of infection. BALB/c mice were immunized orally with 10^8^ or 10^9^ CFU of LT39 (Δ*fnr*Δ*arcA*Δ*fliC*) and boosted after 28 days. The animals were challenged orally with 100 LD50 of ATCC1402 strain 1 month after the booster. The animals were fully protected against challenge with 100 LD50 of the virulent strain when immunized with 10^8^ CFU of the mutant but only 90% when 10^7^ CFU of the mutant was used for immunization. Bacterial organ load in the spleen and liver could also be reduced up to 4 orders upon vaccination with 10^7^ or 10^8^ CFU of the mutant.

The use of LPS components was studied as a live vaccine strategy. Transfer of the whole O-antigene synthetic cluster (*wzx-wbaZ*) from *Salmonella* Newport was tested by Zhang et al. [290] in an *S.* Typhimurium strain. Four combinations were created with a homo- or heterologous expression of the O-antigen on the surface of the recipient *Samonella* Typhimurium strain. BALB/c mice were immunized orally with 1 × 10^9^ CFU of each mutant strain and boosted with the same strain after 4 weeks. One month after the booster, the animals were challenged orally with 100 LD50 of *S.* Typhimurium ATCC14028 or *S.* Newport SLN06 strain. Each of the constructs could protect against challenge with the donor strain when tested at 100 LD50 level. Bacterial load decrease in organs after vaccination was 3–5 orders of magnitude.

### 2.16. Shigella

*Shigella* is a genus of bacteria that is Gram-negative and facultatively anaerobic. There are 4 species: *dysenteriae*, *flexneri*, *boydii*, and *sonnei*. All except the *boydii* species can cause severe dysenteric human disease shigellosis. The bacteria are transmitted via the fecal-oral route and most episodes are associated with poor sanitation and contaminated water. There is no commercial vaccine available [291] despite intensive efforts [292,293] and the only therapy is with antibiotics.

The pathogens have a type III secretion system encoded extrachromosomally on a plasmid. The overall structure of the system is conserved functionally and the system has been a target of different therapeutic approaches [294,295,296,297] due to its role in the infection process [298,299,300,301].

#### Vaccines

The main target in the group is the type III secretion system and 2 of such strategies will be described in the present review. In the first strategy, translocator proteins IpaD-IpaB were used as a vaccine candidate by Chen et al. [302]. The recombinant IpaD-IpaB proteins, separate and a fusion, were purified from *E. coli*, solubilized in a detergent (LDAO or OPOE), and used as antigens in a mouse model of infection with *S. flexneri*. Mice (unspecified) were immunized intranasally with 2.5 μg protein/animal with a double mutant heat-labile enterotoxin (dmLT) and boosted twice over 28 days with the same antigen. Challenge was performed with 1.6 × 10^7^ CFU of *S. flexneri* 2457T strain via intranasal route 28 days after the last booster. The fusion protein could protect up to 90% of the mice from 10^7^ CFU of pathogen delivered intranasally when the LDAO detergent was used for its solubilization during purification and only 50% when the OPOE detergent was used.

In a different approach, Heine et al. [303] used IpaB and IpaD displayed on the surface of *Lactobacillus lactis* bacterial-like particles (BLPs) to immunize mice. The attachment of BLPs was needed to avoid adding detergents to purified proteins for solubilization. BALB/c mice were immunized intranasally with increasing amounts (2.5–53 μg/animal) of proteins alone or coupled to BLPs via the intranasal route and boosted twice over 28 days. Animals were challenged with 6 × 10^7^ CFU of virulent *S. flexneri 2a* 2457T (~11 MLD50) or 1.3 × 10^8^ CFU of S. sonnei 53G (~6 MLD50) orally. The construct protected up to 90% and 80% of animals against the *S. flexneri* and *S. sonnei* challenge, respectively, when immunization was performed with BLPs at the highest dose of antigen. Removal of the BLPs reduced the numbers to 40% and 20% for *S. flexneri* and *S. sonnei*, respectively, for the highest amount of protein used.

### 2.17. Yersinia pestis

*Yersinia pestis* is a Gram-negative facultative anaerobe. The pathogen is the most lethal bacterium known to humans and LD50 for mice in an s.c. model is 1–2 CFU [304]. The pathogen was responsible for the epidemic of the Black Death in medieval Europe and the bacterium is still endemic in Africa, mainly Madagascar, Asia (China, India, Pakistan), South America, and the Western United States [305,306]. There is no vaccine recommended by the WHO despite intensive research [307] and the only treatment is with antibiotics [308].

Genetic analysis of *Y. pestis* genomes identified that the pathogen has a type III secretion system that is important for virulence [309,310]. The system targets macrophages predominantly [311,312] and LcrV plays an important part in this process [313]. There are 2 types of plague: bubonic and pneumonic, the first one transmitted through flea bites [314] and the second—through aerosol. A third form, the septicemic plague, has also been recognized despite being contracted through flea bites like the bubonic plague [315,316]. Clinically, the septicemic plague is the stage where the pathogen is spread systemically and causing sepsis. In a pulmonary model of infection in mice, targeting the tip of the injectisome with anti-LcrV antibodies could efficiently block effector secretion into the cytosol of the host [317].

The capsule protein F1 is also important for phagocytosis [318] and antibodies to it could be detected in sera of convalescent patients [319,320,321]. The most promising vaccine candidates are based on the fusion LcrV-F1 and their combinations [322]. Moore et al. [322] succeded in the stabilization of the F1/V proteins-based formulation for the oral and intradermal delivery, each with a different adjuvant and relatively stable. However, the F1-V combination could be questionable for the F1- mutants of the pathogen [323] and the polymorphism in the LcrV protein that could defeat the F1-V fusion [324,325,326]. Therefore, alternative strategies are being developed (review in [327]).

#### 2.17.1. Secretory System Proteins

The approach is used typically with the F1 capsule protein as it is important for protection. However, separate proteins have been used as well. Swietnicki et al. [328] used purified recombinant YscF needle protein in an s.c. model of *Y. pestis* infection of mice. Swiss Webster mice were inoculated with 20 μg of recombinant protein absorbed with the Ribi adjuvant system via the s.c. route and boosted with the same amount after 30 days. The animals were challenged with 130 LD50 of *Y. pestis* CO92 strain via the same route. Immunization with the recombinant protein could protect 60% of the animals against 130 LD50 of the *Y. pestis* CO_2_ strain.

The YscF protein was also used in protection against a different strain of pathogen that is less virulent. Matson et al. [329] used the recombinant YscF protein to evaluate protection in a mouse model of *Y. pestis* KIM5 strain deficient in the pigmentation locus. Swiss Webster mice were immunized with 40 μg/mouse of emulsified (complete Freund’s adjuvant) antigen via the i.p. route, boosted with the same after 2 weeks using incomplete Freund’s adjuvant and half of the amount after 4 weeks using the incomplete Freund’s adjuvant again. Challenge was performed intravenously via the retro-orbital sinus with 10^1^ to 10^6^ CFU *Y. pestis* KIM5 (*pgm*-) 2 weeks after the last booster. Immunization with the protein could protect against the infection with the attenuated strain as judged by a 130-fold increase in LD50 of the immunized animals versus placebo.

The YopE secreted effector was used in a different vaccine design. Lin et al. [330] used a cholera toxin (CT) fusion with YopE effector fragment (a.a. 69–77) to immunize intranasally mice and then challenge with attenuated *Y. pestis* strains lacking the pigmentation locus (*pgm*^−^) or the virulence plasmid pCD1 (pCD1^−^). C57BL/6 mice were immunized with 1 or 10 μg peptide and 1 μg cholera toxin mix via intranasal route, boosted twice within 21 days, and challenged with 1 × 10^4^ CFU of the D27 strain of *Y. pestis* on days 37 or 56. The protection was proportional to the amount of antigen delivered and could reach 80% as judged by the survival rates. Bacterial loads in the lungs and liver were reduced by 4 logs compared to the placebo.

#### 2.17.2. Live Vaccines

The early attempts to use live vaccines included LcrV antigen delivered by vectors or live attenuated bacteria.

The use of an attenuated strain of *Y. pseudotuberculosis* as a delivery vehicle was described by Singh et al. [331]. The group engineered attenuated *Y. pseudotuberculosis* strains to deliver *Y. pestis* YopE a.a.1-138-LrcV fusion protein. BALB/c mice were immunized orally with ~10^9^ CFU of engineered *Y. pseudotuberculosis* strain and challenged with 10^3^–10^4^ CFU of *Y. pestis* KIM6(pCD1Ap) intranasally (pneumonic plague model) or 2.6 × 10^6^ CFU of the same strain subcutaneously (bubonic plague model) 42 days past the last booster. The strains could protect mice against s.c. challenge with *Y. pestis* strain KIM6 supplemented with *Y. pestis* T3SS-encoding pCD1 selectable plasmid to maintain its stability. The live vaccine could protect up to 60% of mice against 10^6^ CFU of the *Y. pestis* KIM6+ (pCD1Ap) virulent strain in an s.c. model of infection. Intranasal challenge mimicking pneumonic plaque with 10^4^ CFU of the virulent *Y. pestis* KIM6+ (pCD1Ap) was more challenging and the protection varied between 10% and 70% for 2 different constructs. The attenuated strain of *Y. pseudotuberculosis* could also protect mice in an oral model of infection with wt *Y. pseudotuberculosis* PB1+ (1.05 × 10^9^ CFU) and *Y. enterocolitica* WA (2.4 × 10^9^ CFU) strains. The protection was complete for the challenge with 10^9^ CFU of the pathogens.

In another attempt at live vaccine, the capsular antigen caf1(F1) from *Y. pestis* was used. Derbise et al. [173] engineered an attenuated *Y. pseudotuberculosis* VTnF1 strain to deliver caf1 protein from *Y. pestis* into hosts. OF1 or C57BL/6 mice were immunized with the *Y. pseudotuberculosis* VTnF1 strain orally with 10^8^ CFU and challenged intranasally (pneumonic plague) or subcutaneously (bubonic plague) with different amounts of *Y. pestis* 4 weeks or 6 months after immunization. The live vaccine could protect mice from 10^3^ LD50 or more virulent *Y. pestis* CO92 in bubonic (s.c.) or pneumonic (i.n.) challenge. The protection was maintained for 6 months at least although the survival rate would drop to 50% upon challenge.

Delivery of F1-V fusion to the host as a live vaccine was shown by Galen et al. [332]. The group engineered the *Salmonella* Typhi strain to deliver F1-V fusion. The *lcrV* gene was under the control of the OmpC promoter while the *caf1* gene was under the control of the *Salmonella* Pathogenicity Island-2 (SPI-2) promoter. In an animal model of *Y. pestis* CO92 Δ*pgm* attenuated strain i.n. infection, the animals were primed intranasally with 1–2 × 10^9^ CFU of *S.* Typhi vaccines on days 0 and 28, boosted intramuscularly with 0.5 μg on day 42 with recombinant LcrV, and then challenged intranasally with 37 LD50 of the attenuated strain on day 78. The vaccines expressing the F1- LcrV fusion protein with the *caf1M-caf1A-calf1* gene region or only F1 with the stabilization fragment ssb included could offer complete protection of animals as judged by the survival ratios. The protection level dropped to 80% or 50% when LcrV only was used.

Deletion of another secretory system as an attempt to create a live vaccine was used by Bozue et al. [333]. The group deleted the *tatA* gene from the Tat transport system in the *Y. pestis* CO92 strain. Swiss Webster mice were immunized with the deletion strain via intranasal, small particle aerosol, or subcutaneous routes using various amounts of bacteria and observed for survival for 21 days. The survival was generally poor for the i.n. and small particle aerosol challenge but much better for the sub-Q immunization. Attempts for protection from the wt strain infection after mutant immunization were not shown, most likely due to the very weak attenuation of *ΔtatA* mutant in animals.

The importance of the T3SS for *Y. pestis* virulence was demonstrated previously by Swietnicki et al. [310]. Accordingly, Bozue et al. [25] evaluated the live plague vaccine based on the *ΔyscN* gene deletion mutant of *Y. pestis*. Swiss Webster mice were inoculated s.c. with *ΔyscN* mutant at 10^2^–10^7^ CFU per animal and boosted with the same amount after 30 days. Two weeks after the last boost, the animals were challenged with 180 CFU (~90 LD50) of the wt *Y. pestis* CO92. The survival ratio was directly related to the number of mutant bacteria used for immunization and reached 90% for the 10^6^ CFU of the mutant pathogen. The challenge was only with 180 LD50 of the fully virulent *Y. pestis* CO92 strain in the s.c. model of infection in mice.

Deletion of virulence system components was tested in another approach at live vaccine design. Cote et al. [334] removed the T3SS ATPase YscN and compared the mutant to those with the *pgm* locus and the pPst (pPCP1 plasmid) genes removed in the aerosol and s.c. models of mouse infection. Swiss Webster or CD-1 mice were injected via the s.c. route with a single dose of the mutant strain and challenged with a virulent strain via s.c. or aerosol routes after 28 days. In a variation, some animals were vaccinated twice with the pathogen via the s.c. route and challenged with the virulent strain after 28–30 days. For the CD-1 mice, vaccination with 9.4 × 10^6^ CFU of the *Y. pestis* CO92 *ΔyscN* mutant could protect all animals from 478 LD50 of *Y. pestis* CO92 (2.5 × 10^4^ CFU) in the s.c. challenge. In the aerosol challenge, the animals were vaccinated with 1 × 10^7^ CFU of mutant strain and exposed to 26 LD50 of *Y. pestis* CO92 (8.71 × 10^5^ CFU). The protection was only 20% as judged by the survival rates.

Testing on Swiss Webster mice showed that double vaccination with the mutant strain (1.03 × 10^7^ CFU in the first immunization and 0.85 × 10^7^ CFU in the booster) could protect all animals from 316 LD50 (505 CFU) of the *Y. pestis* CO92 strain in the s.c. challenge and 7 LD50 (4.78 × 10^5^ CFU) in the aerosol challenge. However, vaccination with the *Y. pestis* C12 *ΔyscN* mutant lacking the yscN gene and capsule antigen F1 was less successful. In the s.c. challenge model, animals were vaccinated 1.27 × 10^7^ CFU of the C12 mutant and boosted with 0.93 × 10^7^ CFU, and challenged with 316 LD50 (505 CFU) of *Y. pestis* CO92. Double vaccination could protect 40% of the animals while a single vaccination (booster dose) protected only 30% of the animals. In the aerosol challenge mode, even the double vaccination protocol with the C12 mutant could not protect any animals from the challenge.

#### 2.17.3. Outer Membrane Proteins

The use of general bacterial secretions as delivered in outer membrane vesicles for vaccine design was evaluated by Erova et al. [335]. The group evaluated outer membrane proteins Ail (adhesin), Pla (plasminogen activator), and OmpA (porin) versus the F1-V fusion as potential vaccine candidates in the subcutaneous (s.c.) and intranasal (i.n.) models of *Y. pestis* infection in mice. Swiss Webster mice or Brown Norway rats were immunized with 20 μg of antigen via the i.m. route and boosted twice with the same amount every 2 weeks. One week after the last booster, mice were challenged via the s.c. route with 500 LD50 (2.5 × 10^4^ CFU) of the *Y. pestis*
*Δcaf1* [336] mutant (Ail, Omp, and Pla antigens) or 15 LD50 (i.n., 7500 CFU) of the same mutant in the pneumonic plague model. Rats were immunized with 20 μg of F1-V antigen and challenged intranasally with 5 × 10^6^ CFU (10,000 LD50) of the wt strain. For the Ail, Omp, and Pla antigens, rats were immunized with 20 μg of the antigen and challenged via the i.n. route with 4 × 10^3^ to 5 × 10^3^ CFU (8 to 10 LD50) or via the s.c. route with 7 × 10^2^ to 8 × 10^2^ (7 to 8 LD50) of the wt strain.

The OmpA and Ail immunization could protect around half the mice against 500 LD50 of *caf1^−^* mutant in the s.c. model while F1-V could protect all the animals from death. The Pla protein was tested in the i.n. model against only 15 LD50 of the same mutant, showing protection for 60% of animals while the immunization with F1-V fusion could protect all the animals from death. The *calf1^−^* mutation attenuated the pathogen 10^3^–10^4^ times in mice to enable measurements of antibody response to weakly protective antigens in the proposed models. The F1-V protein generated a strong antibody response in both models, while in the i.n. model-the Pla protein was strongly immunogenic. The s.c. model generated a strong antibody response for the Ail protein.

## 3. Discussion

The ability to design a vaccine for any bacterial pathogen *a priori* is still elusive. The methodology to predict putative targets exists but the quality of predictions has to be verified experimentally. In some cases, however, the required targets are known and verified but the vaccine constructs still show a poor performance in terms of protection.

In general, live vaccines could present the full antigenic spectrum of the pathogen and stimulate both parts of the immune system: humoral and cellular. The approach typically offers a long-lasting immunity and is not prone to mutations inactivating the effectiveness of vaccines based on 1–2 antigens or their epitopes. The drawback of such an approach is side effects connected with the use of whole-cell products or live viruses.

Vaccines based on live viruses offer the option of stimulating both parts of the immune system but the limited antigenic repertoire caused by structural limitations of modified viral proteins is their weak point as the vaccines can be easily rendered ineffective by mutations in the pathogen. The other drawbacks associated with live viruses are their side effects connected to the recognition of viral nucleic acids by the innate immune system and the replication cycle. The last feature is also an advantage due to the constant and prolonged stimulation of the immune system by the replicating virus.

Vaccines based on recombinant antigens stimulate the immune system strongly and selectively for a short time but the immune response is frequently biased towards the humoral part. To correct the problem, different adjuvants are used to have the cellular part also engaged and create a long-lasting immunity. The drawback of vaccines based on recombinant antigens is their limited antigenic repertoire that is easily bypassed by mutations in the pathogen. To overcome this limitation, multiple antigens are used which, for practical reasons, are limited to no more than 3 different components in a given vaccine.

The use of Outer Membrane Vesicles is a new technology that offers a broad antigenic repertoire but the quality of preparations may be problematic due to the heterogeneous mix of components. The strategy offers the possibility to bypass mutations in the pathogens inactivating vaccines based on a single component but the technology has not reached a commercial stage.

Approaches based on the carbohydrate coats, LPS and its components, have been in use for a long time. They are typically based on the O-antigen or the whole LPS conjugated to a carrier, a diphtheria toxin or similar, to increase their antigenicity. The vaccines are highly specific and frequently very effective against a given serotype. The drawback of such an approach is the variability of the O-antigen and the LPS outer core necessitating a constant addition of new components specific to a given serotype. The carbohydrate components require isolation from natural sources, frequently biohazardous materials, on a large scale as the technology to make in vivo recombinant O-antigens coupled to the carrier has not reached a commercial stage.

Vaccines based on nucleic acids can be divided into 2 groups: DNA-based and RNA-based. The first group uses plasmids replicating in the host as they are more stable than linear DNA, easier and cheaper to produce than recombinant proteins, and can be used by the host’s machinery directly. The drawback of such an approach is the limited amount of antigen produced, relatively fast elimination of the plasmid by the host, and a limited antigenic repertoire. The RNA-based vaccines use encapsulated RNA delivered locally. The approach allows for engaging the host’s machinery in the production of proteins without the necessity of going through an mRNA intermediate as for the DNA-based vaccines. The vaccines are easy to design and manufacture but are unstable due to the rapid degradation of the RNA component. Therefore, the vaccines require a deep-freeze (−70 °C) storage and offer a limited shelf life, making them difficult to deploy on a large scale without a preexisting logistical network. The vaccines offer a limited antigenic repertoire due to the limitations on the length of the used RNA and can be easily bypassed by mutations in the pathogens. However, they can be very effective as demonstrated for the COVID-19 vaccines.

In the presented review of vaccines for selected bacterial pathogens, most of the published data is focused on constructs offering protection to bacterial challenges not exceeding 100–200 LD50, a level typically assumed to be background when designing commercial vaccines. An increase in the protection level as demonstrated for discussed vaccines was observed for constructs combining other components of the system not belonging to the secretory systems with the knowledge of its vulnerable points (F1-V vaccine for *Y. pestis*) or simply by using engineered/attenuated organisms (Select Agents vaccines against *F. tularensis* and *Y. pestis,* whole cell-based DTwP against *B. pertussis*, live attenuated Ty21 *S.* Typhimurium vaccine, BCG vaccine against *M. tuberculosis*). The strategy to use the full antigen spectrum of the pathogen is far superior to that based on a combination of a few antigens as clearly demonstrated for the emerging cases of whooping cough in the population vaccinated with the acellular DTaP. Since the transition to acellular prophylactics based on potential side effects connected to the engineered live vaccines has been enforced by the regulatory bodies [337], the effectiveness of future vaccines has been degraded and is likely to stay so when using the same products in mass vaccination efforts over a long period. A correction of the biased immune responsible for part of the observed results has been demonstrated [52] but the DTaP vaccine is still administered without the potential correction.

The virulence systems are an attractive target for vaccines but the use of their components only has not been very successful except for the *S. flexneri* vaccine candidate based on translocator proteins IpaB-IpaD attached to bacteria-like particles [302] and the F1-LcrV construct for *Y. pestis* vaccine candidate [338]. The opposite strategy, engineering pathogens to remove virulence systems, was only modestly successful [25,334]. It is possible that combining virulence system components with other bacterial antigens could be a strategy to develop new and effective vaccines. The constructs, however, will offer limited antigenic repertoire when only recombinant proteins are used and could be easily bypassed by mutated pathogens. A much better option would be to use carriers containing empty bacterial shells to increase antigenic repertoire but the cellular components may give rise to undesirable side effects.

## Figures and Tables

**Table 1 biomolecules-11-00892-t001:** List of bacterial pathogens reviewed.

Pathogen	Approved Vaccine	Secretory Systems	Notes
*Acinetobacter baumannii*	No	I, II, VI	
*Bacillus anthracis*	Yes (restricted use)	II (general), Tat	Select Agent
*Bordetella bronchiseptica*	Yes (animals)	I, II, Tat, III, VI	
*Bordetella pertussis*	Yes	I, II, Tat, III	
*Brucella abortus*	Yes (animals)	I, II, Tat, IV	
*Brucella melitensis*	Yes (animals)	I, II, Tat, IV	
*Chlamydia trachomatis*	No	II (general and pathogenic), III	
*Pathogenic E. coli*	Yes (selected variants)	I, II (general and pathogenic), III, VI	Select Agent (EHEC)
*Francisella tularensis*	Yes (restricted use)	I, II (general), VI	Select Agent
*Helicobacter pylori*	No	II (general), Tat, IV, V	
*Legionella pneumophila*	No	I, II (general and pathogenic), IV	
*Mycobacterium tuberculosis*	Yes	II, Tat, VII	BSL-3
*Proteus mirabilis*	No	I, II, III, V, VI	
*Pseudomonas aeruginosa*	No	I, II (general and pathogenic), III, Tat, VI	
*Salmonella enterica*	Yes (selected serovars)	I, II, III, Tat, VI	
*Shigella spp.*	No	I, II (general), III, Tat, VI	
*Yersinia pestis*	Yes (restricted use)	I, II (general and pathogenic), III, VI	Select Agent

## Data Availability

Not applicable.

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
