# Peer review of "Secretory System Components as Potential Prophylactic Targets for Bacterial Pathogens"

_biomolecules, 2021, doi:10.3390/biom11060892_

Round 1
Reviewer 1 Report
This is a well written, very ambitious review of a number of pathogens. Obviously it is imposible to cover all the detail for all the pathogens, but here are 1 or 2 potentially useful additions/suggested amendments:
page 3 : anthrax vaccines. The AVP cannot be described as a cellular vaccine. You have correctly stated that is a suspension of culture filtrate containing predominantly PA, with traces of LF and EF and other proteins, but no cells. See Tapasvi et al. Human Vacc and Immunotherapeutics, 2021, issue 3, vol 17.
Pertussis: the use of too few epitopes is not the only reasin for resistance. The polarisation of the immune sreponse elicited by the acellular vaccine (Th2/th17) differs from that elicited by the whole cell vaccine (Th1) (see e.g. Mills KHG et al)
Francisella-see alos Jia & Horowitz Front Cell Infect Microbiol May 2018
Line 753-754 Plahue is endemic on 3 continents: Africa, Asia and america (predominantly S.America) and aprticuolarly on Madagascar.
Line 804: 'another one with an extra single-strand binding protein' makes no sense
Yes the F1-V fusion has led the field to date, but alternatives are emerging (see Moore et al 2018, vaccine 36, 5210; and Sun and Singh NPJ vaccines 2019, 4,11.
Author Response
Dear Reviewer,
Thank you very much for pointing out deficiencies in the manuscript. The requested references and corrections have been added to the revised manuscript.
Sincerely,
W. Swietnicki, Ph.D.
Reviewer 2 Report
There is a reference to B. abortus in the B. pertussis section And then need to add Das et al 2020 Frontiers in Immunology to the Pseudomonas section.Author Response
Dear Reviewer,
The requested references have been added to the revised manuscript. The author is very thankful to the reviewer for pointing out deficiencies.
Sincerely,
W. Swietnicki, Ph.D.
Reviewer 3 Report
The manuscript attempts to review vaccine strategies for many pathogens, which is an undeniably interesting idea. Therefore, the manuscript must have required a lot of work given the number of pathogens mentioned in the text. However, I have some remarks/suggestions to make about the content and form of the manuscript in order to avoid bias and to make it more attractive, and to make it more focus.
- With respect to the content of the text, it is undeniably informative. However, there are missing data from the literature and there are inaccuracies, which is not surprising given the broad scope of the review. Two examples,
- For pertussis, the authors did not mention a new vaccine (BPZE1) while there is a promising clinical trial (Lin A, J Clin Invest 2020).
- For Y. pestis, the author forgets that plague is mostly endemic in Africa. In addition, the author did not describe the full range of vaccine strategies. The author will find useful information in the "Special Issue Yersinia pestis Biomolecules" MPDI 2021 (If the author uses this information, please cite the original article (laboratory work) and not necessarily the review).
Therefore, I am concerned that there is a truncated presentation of literature data for several pathogens mentioned in the manuscript (for which I am not an expert ). The author may choose to limit the examples of strategies implemented. If so, this should be clearly stated, otherwise the reader will have biased information.
- In fact, I have trouble understanding the exact purpose of the review as it is. From an architectural and stylistic standpoint, it was difficult for me to understand why the author emphasizes the secretory apparatus because the vaccine aspect sometimes appears detached from this information. In fact, this confusion made me wonder if the author should not briefly present the secretion systems and the control strategies in place. The information given on the pathogens is reduced, maybe even too much. My feeling comes from the fact that the author focuses a lot on the secretion systems, and the introduction goes in that direction. Perhaps my suggestion would avoid the feeling of repetition. In this way, it might also clarify the purpose of the review.
- With regard to the architecture, sometimes, the text appears to me as a succession of information without necessarily a direct link. At some point, I had the feeling of having a catalog, a telegraphic style. Even if this is very useful, I suggest to the author to modify the text to avoid this feeling and to make it more attractive.
- In case the author prefers to keep his architecture (it is his right of presentation of course), I find little use to put titles for paragraphs that are limited to 1 or 2 lines of text. In fact, the paragraphs "virulence systems" should be renamed because not all factors of the system are mentioned. In addition, the amount of information related to virulence systems and/or vaccination is disproportionate depending on the bacterium considered. Sometimes there may be too much or too little information. For example, doses and routes of administration are given in one case but not in another. I suggest that the author try to harmonize the text while keeping in mind that too much information may hinder the reader.
- Lastly, the limitations of each vaccine approach are not presented. For example, it is clear that some vaccines could be defeated by polymorphisms or the fact that some strains do not express the targeted antigen, or even can be selected because of vaccination. I find that a paragraph mentioning these limitations (with associated references) is useful in the discussion. In fact, reference 266 is inadequate to support the points made.
Other comments
- The introduction has many statements but only 3 references (two of which, 2 and 3, refer to the same pathogen). This is also the case in the discussion. The author must provide references for each statement or else it is the author's opinion. In the latter case, the author must write that it is his/her opinion.
- Line 41, I am surprised to read "vaccine design strategies ignore the biology of the pathogen" because there are many examples against this statement. In fact, lines 60-61 contradict line 41.
- Lines 32-36. It seems inappropriate to me to say that the slow progress in the development of drugs favors the concept of antivirulence. In my view, it is not the slow pace of investment but the idea of finding new and innovative therapeutic targets to combat multidrug resistance to antibiotics that has led to the concept of antivirulence, or at least has given rise to this concept.
Lines 44-45. The statement that "The strategy has to change soon if we are to expect a substantial breakthrough in novel vaccines in the future" is unclear to me. The information given to support this statement is confusing. In fact, I wonder if the ability of humans to have produced a vaccine against covid19 in record time contradicts this statement.
Line 53: Bacterial transport systems are not pathogen-specific. Nor are some secretory systems. Therefore, I suggest to modify the sentence accordingly
Lines 839-844, the sentence is unclear.
Finally, it is not normal that the name of the bacteria is not in italics.
Author Response
Dear Reviewer,
Thank you very much for the comprehensive and insightful review of the manuscript. All questions have been answered and the manuscript corrected, mostly according to the suggestions.
Sincerely,
W. Swietnicki, Ph.D.

Round 2
Reviewer 3 Report
The author has made substantial changes.
- I have 2 general comments that give my point of view as a reader. The author can take them into account or not.
- I also have a response to a comment that was misinterpreted and seems to have hurt the author, which was obviously not my intention.
- I have comments that need to be addressed prior to publication regarding pestis.
My 2 general comments that may or may not be taken into account
Regarding the response to my comments 1 and 2. I did understand that the focus of the review was on secretory systems, and I agree that it is a good idea to omit other vaccine strategies that do not fit the manuscript (response to my comment #1). However, the author responds below that "other strategies are also added to provide a broader view of the vaccine field for certain pathogens." I still think this undermines the main purpose of the manuscript. However, I respect the author's viewpoint and choice.
Regarding the response to my comments #4. The author decided to respond to my suggestion of harmonization by adding a lot of information/details. I should have pointed out that he really needed to keep in mind that too much information could be distracting to the reader. That said, it is the author's point of view and I respect his choice.
My clarification on my previous comment to avoid any misunderstanding
I agree that author's opinion is not necessarily an accident. It is evidence of a thought. When the author's thought is supported by facts, the community accepts it better. Also, citing those who have already contributed is a form of respect. My comment is along these lines and is not intended to hurt your feelings!
Comments that should be addressed
- Line 1264. There are not 2 but 3 forms of plague. The third is septicemic plague, which can be contracted by various means, including bites from infected fleas. (Flexner S. Am. J. Med. Sci 1901 ; Crowell B. C. (1915) Philippine J. Sci ; Hull, J. Infect. Dis 1987 ; Sebbane Proc Natl Acad Sci USA 2006).
- Line 1274. The author wrote : « F1-V combination could be questionable for the F1- mutants of the pathogen [321] ». It should be noted that this vaccine could be defeated by a polymorphism in LcrV. (Motin Infect Immun. (1994) ; Roggenkamp Infect Immun. (1997) ; Daniel Vaccine. (2009) ; Daniel. Front Immunol (2019).) Since LcrV is part of a secretory system that is the subject of this review, I suggest mentioning this important point.
Author Response
The Author is very grateful to the Reviewer for constructive criticism. Some of the remarks were probably understood differently than intended and led to extra work. The others were answered.
- The form of septicemic plague as a separate form of Y. pestis infection has been acknowledged and 2 references have been added. The other 2 references could not be reached due to their age and were omitted.
- References about a mechanism to defeat F1-V fusion based on the polymorphism of the LcrV protein are very important. Proper references were added as requested. The reference of Daniel et al. from 2009 does not show polymorphism of LcrV in the experiments presented and has been omitted. In contrast, the reference of Daniel et al. from 2019 does show polymorphism of LcrV as the causative factor of the inefficacy of strategies based on LcrV vaccines and has been added as requested.
Again, the Author would like to thank the Reviewer for the important corrections.